# Canopy structure modulates the sensitivity of subalpine forest stands to interannual snowpack and precipitation variability

Max Berkelhammer[a], Gerald F Page[b,c], Frank Zurek[a], Christopher Still[c,d], Mariah S Carbone[e,d], William Talavera[a], Lauren Hildebrand[a], James Byron[a], Kyle Inthabandith[a], Angellica Kucinski[a], Melissa Carlson[a], Kelsey Foss[a], Wendy Brown[d], Rosemary WH Carroll[f], Austin Simonpietri[e], Marshall Worsham[g], Ian Breckheimer[h,d], Anna Ryken[i], Reed Maxwell[j,k], David Gochis[l], Mark Raleigh[m], Eric Small[n], and Kenneth H Williams[o,d]

[a]Department of Earth and Environmental Sciences, University of Illinois Chicago, 845 W Taylor St. Chicago, IL 60607
[b]School of Environmental and Conservation Sciences, Murdoch University, 90 South St, Murdoch, Western Australia, 6150
[c]Forest Ecosystems and Society, Oregon State University, Corvallis, OR, 97331 USA
[d]Rocky Mountain Biological Laboratory, Crested Butte, CO
[e]Center for Ecosystem Science and Society, Northern Arizona University, Flagstaff, AZ, USA
[f]Division of Hydrologic Sciences, Desert Research Institute, Reno, NV
[g]Energy and Resources Group, University of California, Berkeley, CA, USA
[h]Western Colorado University, Gunnison, CO
[i]Hydrologic Science and Engineering, Colorado School of Mines, Golden, Colorado, USA
[j]High Meadows Environmental Institute, Princeton University, Princeton, NJ 08544, United States of America
[l]Research Applications Laboratory, National Center for Atmospheric Research, Boulder, Colorado, USA
[k]Department of Civil and Environmental Engineering, Princeton University, Princeton, NJ 08544, United States of America
[m]College of Earth, Ocean, and Atmospheric Sciences, Oregon State University, Corvallis, OR, 97331, USA
[n]Department of Geological Sciences, CU Boulder, Boulder CO
[o]Lawrence Berkeley National Laboratory, Berkeley, CA, USA

**Correspondence:** Max Berkelhammer (berkelha@uic.edu)

**Abstract.** Declining spring snowpack is expected to have widespread effects on montane and subalpine forests in western North America and across the globe. The way tree water demands respond to this change will have important impacts on forest health and downstream water subsidies. Here, we present data from a network of sap velocity sensors and xylem water isotope measurements from three common tree species (*Picea engelmannii*, *Abies lasiocarpa* and *Populus tremuloides*) across a hillslope transect in a subalpine watershed in the Upper Colorado River Basin. We use these data to compare tree- and stand-level responses to the historically high spring snowpack but low summer rainfall of 2019 against the low spring snowpack but high summer rains of 2021 and 2022. From the sap velocity data we found that only 40% of the trees showed an increase in cumulative transpiration in response to the large snowpack year (2019), illustrating the absence of a common response to interannual spring snowpack variability. The trees that increased water use during the large snow year were all found in dense canopy stands - irrespective of species - while trees in open canopy stands were more reliant on summer rains and thus more active during the years with modest snow and higher summer rains. Using the sap velocity data along with supporting measurements of soil moisture and snow depth, we propose three mechanisms that lead to stand density modulating the tree-level response to changing seasonality of precipitation. (1) Topographically-mediated convergence zones have consistent access

to recharge from snowmelt which supports denser stands that are more reliant and sensitive to changing snow. (2) Interception of summer rain in dense stands reduces throughfall of summer rain to surface soils limiting the sensitivity of the dense stands to changes in summer rain. (3) Shading in dense stands allows the snowpack to persist deeper into the growing season providing locally high reliance on snow during the fore summer drought period. Combining data generated from natural gradients in stand density, like this experiment, with results from controlled forest thinning experiments can be used to develop a better understanding of the possible responses to proposed low-snow futures in subalpine ecosystems.

## 1  Introduction

Across the mountainous regions of the western US there has been a widespread decline in spring snowpack (Mote et al., 2018). The spring pulse of snowmelt recharges deep soil layers providing a water source for forest ecosystems that can persist through the fore summer drought period (Harpold and Molotch, 2015; Wainwright et al., 2020; Sloat et al., 2015; Coulthard et al., 2021). Isotopic and modeling studies have consistently shown that winter precipitation continues to act as the primary water source for subalpine trees for a period of months after the snowmelt pulse (Berkelhammer et al., 2020; Allen et al., 2019; Martin et al., 2018; Love et al., 2019; Kerhoulas et al., 2013). A reduction in the magnitude and duration of snowmelt inputs combined with higher warming-induced increases in evaporative demand have been shown to manifest in these forests as phenological shifts, increases in mortality, thinning, crown dieback and greater susceptibility to disturbance (Kelsey et al., 2021; Allen et al., 2010; Knowles et al., 2023; Carrer et al., 2023; Cooper et al., 2020).

Despite the broad ecological importance of snowmelt for subalpine forests (Trujillo et al., 2012), utilization of summer rain is also critical to the functioning of these ecosystems, particularly during periods with reduced snowmelt inputs (Berkelhammer et al., 2017; Strange et al., 2023). For example, aspen (*Populus tremuloides*) rely on shallow soil moisture replenished by summer rains to alleviate periods of drought stress (Anderegg et al., 2013). In contrast, subalpine conifers such as trees from *picea*, *abies* and *pinus* genus have generally been shown to be less responsive and reliant on summer precipitation (Pataki et al., 2000). Although summer rain only contributes 10-20% of annual precipitation inputs for many areas in the western US, these modest precipitation inputs are sufficient to increase soil matric potentials above thresholds that can cause hydraulic damage to transpiring trees. This facilitates active water uptake by trees even during dry periods late in the summer and into the fall (Samuels-Crow et al., 2023). Furthermore, the convective storm systems associated with summer rain locally increase humidity and moisten the surface soils, thus reducing evaporative demand on trees (Strange et al., 2023). While a lot of attention has been given to declining trends in spring snowpack (Schmitt et al., 2024), summer rain in the western US may also be experiencing a decrease or possibly a change in frequency and intensity though any persistent or spatially coherent trends remain less obvious (Holden et al., 2018; Pascale et al., 2017).

To understand the response of subalpine forest systems to changes in seasonal precipitation inputs, a better understanding of the factors that influence the seasonal origins of water used by trees is needed. Previous work in subalpine forests has often

considered this problem from the perspective of species traits and their impact on the timing and depth trees extract their water from. For example, Grossiord et al. (2017) showed that co-located piñon and juniper had opposing belowground responses to water stress manifesting in piñon trees seeking out deeper winter-sourced water during drought. Aboveground traits also influence water use patterns such that species like *populus tremuloides* with lower susceptibility to embolism can maintain high rates of transpiration during drier periods late in the summer (Pataki et al., 2000). On the other hand, species that have a higher susceptibility to embolism from air seeding such as *abies lasiocarpa* face greater risk of extracting water from desiccated soils and during periods with high vapor pressure deficits (VPD). Consequently, they are less likely to remain active during periods of high evaporative demand and low soil moisture that precede or occur after the emergence of periodic summer rains. These species thus have a tendency for higher relative reliance on snow because the risk of transpiring during periods of low soil moisture outweighs the benefits of access to sporadic rain events (Berdanier and Clark, 2018). In addition, differences in species-level allocation to leaf area and variations in phenology affect interception of snow and rain as well as surface radiation loads to the surface, which influence sublimation of snow and soil evaporation. For example, results from subalpine forests in Colorado show that aspen stands may experience 40% higher effective precipitation rates (Thomas, 2016) and 20% higher rates of winter sublimation then nearby conifer stands (LaMalfa and Ryle, 2008). These processes effectively shift what seasonal precipitation inputs are present in the surface soils of forest stands composed of different species composition.

To study the question of seasonal water utilization strictly in terms of species-level traits has limitations because the distribution of species within a watershed is generally sensitive to hillslope position (Metzen et al., 2019). Thus, species may experience different temperatures and vapor pressure deficit, depth to groundwater, snowpack buildup, timing of snow disappearance and radiation loading all of which may influence preferential utilization of seasonal water sources (Brooks et al., 2015; Martin et al., 2018; Molotch et al., 2009; Fabiani et al., 2022; Cooper et al., 2020). For example, using sap velocity and water isotope data from a sub-alpine watershed in Montana, Martin et al. (2018) found a measurable increase in reliance on snowmelt by subalpine fir over a 350 m elevational gradient. They argue that fir trees in the lower elevation plots rely on more snowmelt due to shallower groundwater downslope and convergence of lateral near-surface flow, both of which lead to persistent access to snow melt waters through the growing season. The lower elevation snow-reliant plots are thus more sensitive to annual snowmelt inputs and more responsive to interannual shifts in the snowpack. The tree water use pattern observed by Martin et al. (2018) supports a view of hillslope ecohydrology where vegetation near the bottom of the hillslope or in local topographically-mediated convergence zones are more connected to changes in snowmelt inputs (Hoylman et al., 2018; Graup et al., 2022). From this perspective, the species-level traits may be less important to determining the seasonal origins of water use than the position on the hillslope where they proliferate.

In addition to species traits and hillslope position, stand density generates myraid effects on water pathways through the soil-plant-atmosphere continuum including modulating interception of rain and snow in the canopy, competition for soil water between plants and changes in soil infiltration (Tague et al., 2019). Because stand density can be actively managed, changing density may be used to reduce vulnerability to ongoing changes in climate and snow hydrology (O'Donnell et al., 2021;

Belmonte et al., 2022). Consequently, the effects of stand density on seasonal water access and water stress have been studied extensively through thinning experiments (e.g. Bréda et al. (1995); Kerhoulas et al. (2013)). One of the most consistent effects of stand density in subalpine forests is that thinned stands tend to experience a larger buildup of winter snowpacks due to reduced canopy interception. However, this effect does not always translate to more water available to plants because higher radiative inputs to the surface can generate melt that occurs earlier in the spring and ahead of the period of most active water use by trees (O'Donnell et al., 2021). The thinned stands also experience higher throughfall of summer rain on the order of 10-20% with a 50% reduction in basal area index (Thomas, 2016; Mazza et al., 2011). In addition to reduced inputs of precipitation in unthinned stands, there is also increased levels of competition for water such that individual trees in unthinned stands experience higher levels of water stress (Bréda et al., 1995) while trees in thinned stand tend to compete with shallow rooted understory plants that emerge in canopy openings (Kerhoulas et al., 2013). Lastly, a number of studies have also shown that stand density influences soil properties such that the hydraulic conductivity and infiltration rate of soils is higher in unthinned stands which can actually increase available soil water despite higher levels of canopy interception (LaMalfa and Ryle, 2008; O'Donnell et al., 2021; Belmonte et al., 2022).

The cumulative effect of thinning is that dense (i.e. unthinned) stands experience drier surface soils, higher levels of water stress and reduced tree-level transpiration (Tague et al., 2019). Kerhoulas et al. (2013) showed that not only does thinning increase total water availability to trees but this change also shifts the seasonality of water sources towards an increase in reliance on snowmelt and winter precipitation. The higher preferential use of winter precipitation in thinned stands is hypothesized to arise from a combination of decreased interception of snow, the presence of taller and deeper rooted trees and an increase in shallow-rooted understory vegetation that compete for summer rain inputs in the shallow soil layers. An increased reliance on winter precipitation in thinned *pinus ponderosa* stands was also found through analysis of the isotopic composition of tree rings by Sohn et al. (2014). However, other studies such as Fernandes et al. (2016) suggest an opposite effect where thinned stands rely more heavily on summer rain due to higher levels of through-falling rain and higher losses of snowpack to runoff and sublimation. The use of thinning experiments document myriad ways stand density could affect reliance and/or sensitivity to changes in precipitation seasonality. However, in the context of unamanged forests where hillslope position, stand density and species distribution co-vary, the competing effects of soil properties, subsurface flow and traits make it more difficult to predict reliance on and sensitivity to changing snow. For example, the location of dense stands often indicates the presence of shallow groundwater recharged by snowmelt (i.e. a high reliance in snowmelt) despite the fact that these stands may also lose more of the potential incoming snow to interception and sublimation.

In this study, we present an interannual analysis of transpiration fluxes (via sap velocity sensors) and water sources (via stable isotope analysis) from a network of sites along a ∼500 m hillslope transect in the Upper Colorado River Basin that includes a mixture of fir, spruce and aspen that are typical for this region. The instrumented and sampled stands include different combinations of co-existing species and fall into a cluster of more open stands - that are typical of this hillslope - and a cluster of dense stands that were in localized convergence zones. We take advantage of the sensors operating across years

with opposing seasonal precipitation inputs (high snow and low summer rain vs. low snow and higher summer rain) to test how species, hillslope position and stand properties influence the response of trees to interannual changes in precipitation seasonality. We document differences in water sources across species and show that trees in the denser canopy stands were
120 more negatively affected by the low snow year. From these results we provide hypotheses to explain how stand density might influence sensitivity to interannual snow inputs in an unmanaged watershed and how this result contrasts with those generated from controlled thinning experiments.

## 2 Methods

### 2.1 Sap Velocity network

We installed a network of sap velocity sensors at six sites across a hillslope transect on Snodgrass Mountain outside of Crested Butte, CO in the Upper Colorado River Basin (Figure 1) (Fuchs et al., 2017). The sites span a 500 m range in elevation transitioning from a dominance of Trembling aspen (*populus tremuloides*) at the lowest elevation, to a mixture of Engelmann spruce (*picea engelmannii*) and subalpine fir (*abies lasiocarpa*) towards the top. At each site, two mature and visibly healthy
trees of each species were selected for instrumentation. Because sites had between 1-3 species present, each stand had between 2-6 instrumented trees. To measure sap velocity, we used the SFM1 sensor manufactured by ICT International, Armidale, NSW, Australia. The sensor logs an estimate of sap velocity using the heat ratio method from Burgess et al. (2001) and briefly reviewed here. Each sensor includes three stainless steel probes of 1.3 mm diameter and 35 mm length. The probes are installed in the tree by removing the outer tree bark and drilling three parallel holes spaced precisely 15 mm apart using a drill guide.
Care was taken to drill the holes as a series of small incremental steps with a low drill speed to minimize wounding that affects the conductivity of wood around the probes. The middle probe includes a 12 VDC heater that is powered on for ∼5 seconds when a measurement is made. The upper and lower probes include thermistors with a resolution of $0.001^oC$ positioned at 7.5mm and 22.5 mm along the probes in order to estimate sap velocity at two depths in the sapwood. We programmed our sensors to turn on and make a measurement every ∼15 minutes though this interval was reduced for some trees where radiation
loads on the solar panels were limited. To estimate sap velocity we use Equations 1-3 below.

$$V_h = \frac{k}{x}\left[ln\frac{t_1}{t_2}\right]3600 \tag{1}$$

$V_h$ is the uncorrected heat pulse velocity, $k$ is the thermal diffusivity of wood, $x$ is the distance between the heater and thermistors (15 mm), $t_1$ and $t_2$ are the increase in temperature measured at the two equidistant points above and below the heater where the thermistors are installed. This measurement then has to be corrected using a wounding coefficient that accounts for
the reduction in the conductivity of wood once it has been wounded during the installation process. To estimate the wounding effect, we use the approach outlined in the sensor manual that is based on earlier work by Swanson and Whitfield (1981) and described by Equation 2 below.

$$V_c = bV_h + cV_h^2 + dV_h^3 \qquad (2)$$

$V_c$ is the corrected heat pulse velocity, $V_h$ is the uncorrected heat pulse velocity from Equation 1 and *a,b,c* are a series of empirically derived wounding coefficients that vary based on the size of the wound which we measured each year when the probes were reinstalled. Finally, we estimate the sap velocity using Equation 3 which solves for the velocity of water within the wood matrix based on Marshall (1958).

$$V_s = V_c \rho_b \frac{c_w + m_c c_s}{\rho_s c_s} \qquad (3)$$

$V_s$ is the sap velocity in units of $\frac{cm}{h}$, $V_c$ is the corrected heat pulse velocity from Equation 2, $\rho_b$ and $\rho_s$ are the densities of wood and water, $c_w$ and $c_s$ are the specific heat capacities of wood and water and $m_c$ is the water content of sapwood which was measured for each tree.

Sensors were installed on the northern side of each tree in June 2019 and re-positioned on the same tree each spring until 2022, excluding 2020 when field work was unable to be conducted due to COVID-19. Bark-depth, sapwood depth, sapwood density and water content were measured in July 2019, with wound diameter also determined following each re-installation. Sapwood depth was determined on two 5 mm diameters cores taken on the northern and southern sides of the tree at 1.2 m height using an increment borer (Haglof, Langsele, Sweden) and stained with methyl orange to indicate active sapwood. An average sapwood depth was calculated for each tree. Although the sensors ran during 2020, data from this season was not included in these analyses owing to observations that wounding effects were too severe to keep probes in the same location for multiple seasons and still produce reliable estimates of sap velocity. Heat-pulse velocity measurements were restricted on some trees to the daytime only - when sap velocity typically peaks - to reduce power consumption overnight. Consequently, nighttime data were limited and the focus of our analysis is on midday patterns. As discussed in Section 3.1, one focus of the analysis of sap velocity was on differences between 2019 and 2021/2022. To do this, we calculated weekly averaged daytime sap velocity and subtracted each weekly value from 2019 against the averaged weekly value from 2021 and 2022. We did not include Site #6 in the interannual analysis because there was insufficient data from that site to estimate seasonally-averaged interannual differences in sap velocity.

## 2.2 Water isotope data

During the growing season, approximately weekly measurements of twig water isotopes were made for each of the trees instrumented with sap velocity sensors. Live twigs of ∼3-5 cm diameter were sampled from the trees, debarked, placed in a sealed bag and frozen as quickly as possible. Water from the twigs was extracted cryogenically using a batch distillation method (Berkelhammer et al., 2020). Surface soil samples were collected from a 10 cm depth periodically from each sap velocity site

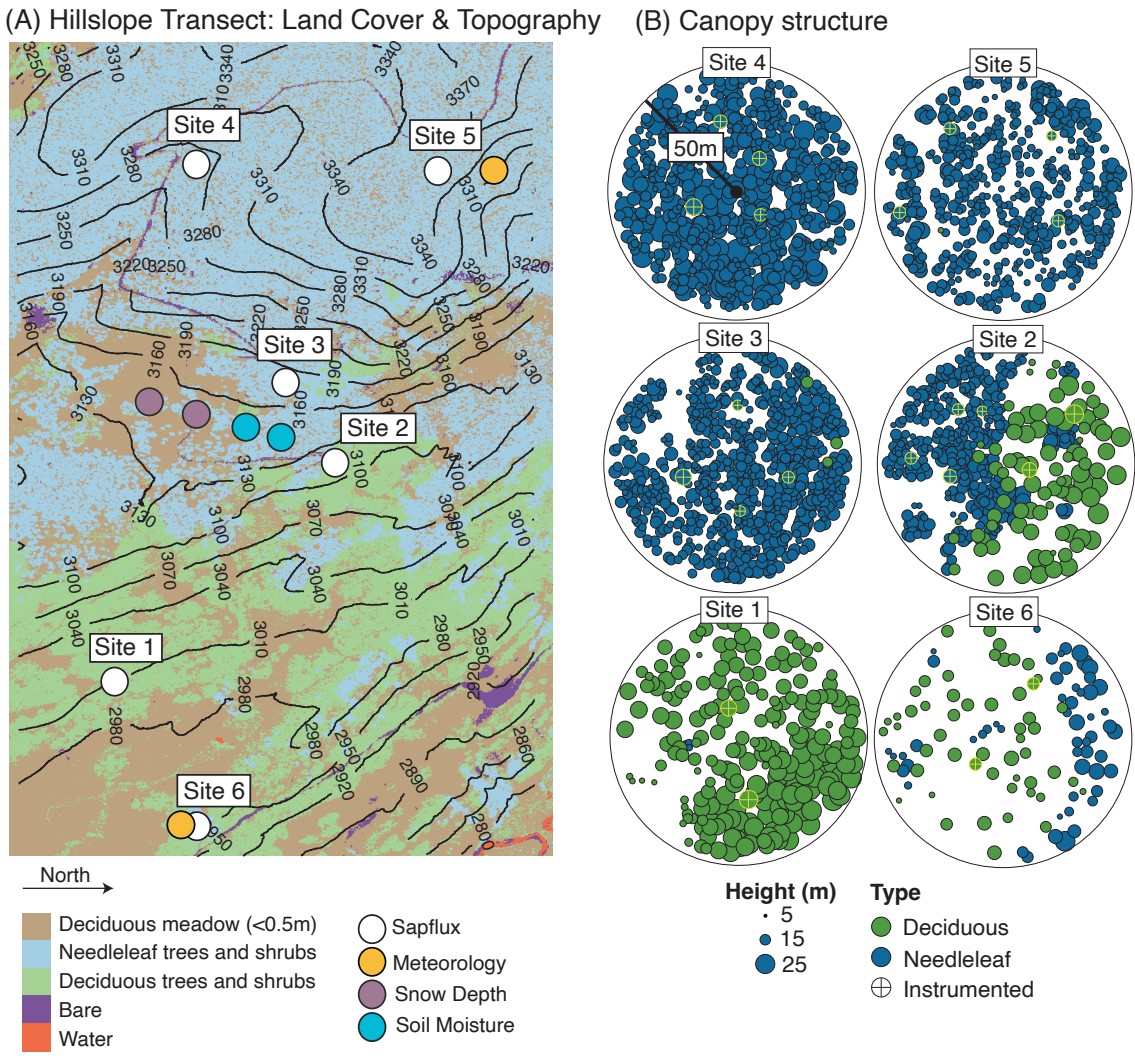

**Figure 1.** (A) Map of the Snodgrass hillslope showing the six sap velocity clusters and multiple meteorological towers projected over elevation and land cover type. (B) Maps of canopy density and tree height for the six plots shown in Panel A. Plots were delineated as 50m radius circle drawn from the center point of the instrumented trees. See Methods for how tree height and individual crowns were delineated.

during the field seasons. The soil samples were extracted following the same method as the twig samples. All extracted soil and twig water samples were analyzed for $\delta^{18}O$ and $\delta^2H$ on a Picarro l2140-i analyzer following the methodology described in Berkelhammer et al. (2020).

To understand the source of water within the extracted twig pool, we utilized additional information on meteoric water inputs to the hillslope. We took advantage of a wide range of isotopic data from this watershed on groundwater, precipitation and snowpack samples which have previously been presented (Carroll et al., 2022a). Twig and soil water samples are subject to evaporative enrichment relative to the precipitation inputs, leading to higher values of $\delta^{18}O$ and $\delta^2H$ as well as a shallower slope between the two isotopes relative to meteoric inputs. To estimate the sources of water in twigs and soils, the measurements need to be projected back to the meteoric water line. We do this following Benettin et al. (2018) and Allen et al. (2019) where an evaporation line is estimated for each sample using measured meteorological data and we solve for the intersection between each sample's evaporation line and the meteoric water line defined by the isotopic ratio of precipitation samples. Lastly, we were cognizant of the possibility for fractionation between $\delta^2H$ of the source water and the twig water that has been noted in a number of recent studies (Chen et al., 2020; Barbeta et al., 2019). This offset has been linked to numerous processes and, according to Diao et al. (2022), is more severe when small samples are used. We attempted to minimize this issue by using large samples that yielded generally greater than 2 ml of water. Nonetheless, we assessed this potential bias by comparing the isotopic ratio of the soil and twig water samples during the early period of the summer immediately following snowmelt when we assume soil water would be minimally evaporated and there would be a nearly homogenous profile in the isotopic ratio of the soil water. During this period, we observed statistically similar values for twig and soil $\delta^{18}O$ but an offset between twig and soil $\delta^2H$ of ∼-6‰ (Figure S2). This negative offset is similar to that observed in previous studies (Barbeta et al., 2019; Diao et al., 2022) and we therefore apply this value as a correction to all measured stem $\delta^2H$ values. Hereafter, we refer to the *twig water* as the estimated isotopic ratio of the source water value following all corrections based on the aforementioned effects. Raw isotopic data are provided in the dataset associated with this publication.

### 2.2.1 Isotopic mixing model

To assess the relative proportion of snowmelt in the twig water samples, we developed a mixing model with three distinct end members. One end member was snowmelt water, whose value was estimated from snowpack and snowmelt measurements with a modest correction for lapse rate across the elevational gradient of our sites (Carroll et al., 2022a). As groundwater from nearby wells in the area was typically hard to distinguish from snowmelt without an additional tracer, we do not attempt to separate the current year's snowmelt from older snowmelt that had recharged the groundwater in previous seasons. The second end member is the weighted average of precipitation during the growing season up until the time of sampling. This was developed by combining the date and amount of each rain event with values for the isotopic ratio of that rain event. The third end member is the isotopic value of the most recent rainfall event. The justification to separate summer rain into two end members was based on the fact that, on the one hand, growing season rain has a cumulative impact that aggregates in the soil horizon while, on the other hand, the most recent rain event may be present in the near surface soils and immediately taken up by the trees. Early in the growing season, these two precipitation end members are nearly identical but they became distinct later in the growing season as the most recent rainfall event tends to be more enriched than the cumulative inputs (Figure S1). We did not consider travel time and storage within the trees which is a limitation to this mixing model in instances when the most recent rainfall event occurred just days before sampling and therefore would not likely yet be present in the sampled twig

reservoir (Knighton et al., 2020). More sophisticated and higher resolution sampling approaches would be needed to resolve these dynamics (Seeger and Weiler, 2021).

We implemented the mixing model using a Bayesian Monte Carlo approach developed and described by Arendt et al. (2015). The principle of this method is that distributions of possible end member mixtures are generated randomly that must adhere to the condition where the fractional contribution of each end member sums to 1 (i.e. *priors*). Samples from these prior distribution are selected using a uniform random walk Monte Carlo simulation. Samples from the simulated end member mixtures are rejected or retained based on their likelihood giving rise to the posterior distribution that provides the most likely mixing model and an estimate of the uncertainty around this model. In developing the prior distributions, uncertainty of each end-member must be defined and we assume that uncertainty is normally distributed around a standard deviation derived independently from our observations. We assume the uncertainty is the same for each measurement and make no specific assumptions about prior distributions being different between species, across sites nor over time. We assume the standard deviation on the twig water measurements are 1‰ for $\delta^{18}O$ and 8‰ for $\delta^2 H$ based on repeat measurements from samples of the same tree. These uncertainties are about an order of magnitude larger than analytical uncertainty and represent the cumulative effects of within tree heterogeneity, sampling, storage/transport and cryogenic extraction. We estimate similar levels of uncertainty for the values of the precipitation and snowmelt end members based on ranges that emerged from the simulations and measurements of snowmelt and precipitation observed in previous studies (Carroll et al., 2022b; Anderson et al., 2016). The bayesian mixing model is similar in function and form as other recent stable water isotopic tree source water studies such as Samuels-Crow et al. (2023).

## 2.3 Meteorological data

We utilized numerous datasets of atmospheric and surface weather data for the analyses and interpretations described in the Results section below. There are 6 meteorological stations on the Snodgrass hillslope transect where the sap velocity measurements were made (Figure 1). These stations - run by three different research groups (i.e. (Simonpietri and Carbone, 2023; Ryken, 2021; Bonner et al., 2022)) - have collected data over different periods of time and have distinct combinations of sensors. Because these datasets have all been published and are publicly available, we refer the reader to the original sources (listed above) for details on the collection approaches. We averaged these datasets to develop a daily mean temperature and humidity dataset for the hillslope to estimate the evaporative mixing lines described above (2.2.1). To do this, we simply took the daily average temperature based on all available measurements for a given day. We utilized snow depth and snow water equivalent (SWE) measurements from a pair of adjacent forest and meadow stations that were derived using both continuous snow height sensors and periodic snow pits (Bonner et al., 2022). This data was used to estimate how the presence of the canopy influenced the timing of snowmelt inputs to the soil. We took advantage of daily precipitation data for weighting the importance of each precipitation event onto the growing season isotopic inputs for the mixing model (Sections 2.2.1 and 2.2.2). We used continuous volumetric water content estimates at three soil depths (5, 15 and 50 cm) from adjacent conifer and aspen sites to illustrate differences in growing season (MJJAS) rain infiltration between the different canopy types (Carbone et al., 2023). Lastly, we

utilized the long term record of rainfall and snowfall from the nearby Gothic, CO weather station (Faybishenko et al., 2023) to place the years of this study into a climatological context.

## 2.4   Remote sensing and GIS

Canopy structure for each site instrumented for sap velocity was measured using airborne LiDAR scanning (ALS), using the 1-m canopy height model (CHM) and vegetation type data (Goulden et al., 2020). We identified the center point of each cluster of
sap velocity probes and captured all trees within a 50 m radius around the center point and classified this tree cluster as a *site* or *stand*. For each site/stand, we identified all the tree crowns using the method of Parkan and Tuia (2018) and identified whether the crown was deciduous (aspen) or coniferous (subalpine fir, Engelmann spruce, or lodgepole pine) based on the vegetation type classification dataset from Goulden et al. (2020). We used previously established allometric relationships between tree height and DBH for these tree species in this region (Hulshof et al., 2015) to estimate the DBH of each crown and then
estimated the sapwood area by using our paired measurement of sapwood depth and DBH by fitting a power-law model with the same order as described by Mitra et al. (2020). We then divided total sapwood area (aspen and conifers) by ground area (fixed at the 50 m radius) to generate a sapwood to ground area estimate for each site. Lastly, we used a local digital elevational model (Goulden et al., 2020) to estimate the topographic position index (TPI) for each site.

## 3   Results

### 3.1   Sap Velocity Data

Instantaneous sap velocity values displayed diurnal and seasonal cycles for all species that followed expectations based on previous work from a similar montane forest system in the region (Pataki et al., 2000) (Figure 1). Both conifer species had peak values early in June during all years but exhibited substantial interannual variations later in the season, such that transpiration rates dropped to very low levels in August during 2019 but persisted measurably into September and October during both 2021
and 2022. These differences reflect the presence of significant summer rainstorms in 2021/2022 relative to 2019. Due to limited early season site access, we did not generate data to capture the early season sap velocity by the conifers. Based on the fact that the species displayed nearly peak rates of sap velocity by early June (i.e. day of year 160), we suspect the early growing season water use was not trivial and is a critical absence in terms of our capacity to close the transpiration water budget. Previous sap velocity work from a subalpine system in the Sierra Nevada of California also suggest that transpiration is active by April and
can reach nearly peak values by mid to late May (Cooper et al., 2020). Although the conifers displayed similar average sap velocity values and temporal patterns, we do note measurable differences in the behavior between species. For example, the sap velocity of subalpine fir relative to Engelmann spruce was shifted earlier in the season and showed higher water use during early morning and evening (though with similar midday peak values for both species). In contrast to the conifers, aspen transpiration began typically in early June following leaf-out with a protracted period of high transpiration that extended into mid
August before showing measurable declines. Transpiration stopped by mid September ahead of leaf senescence. Despite the

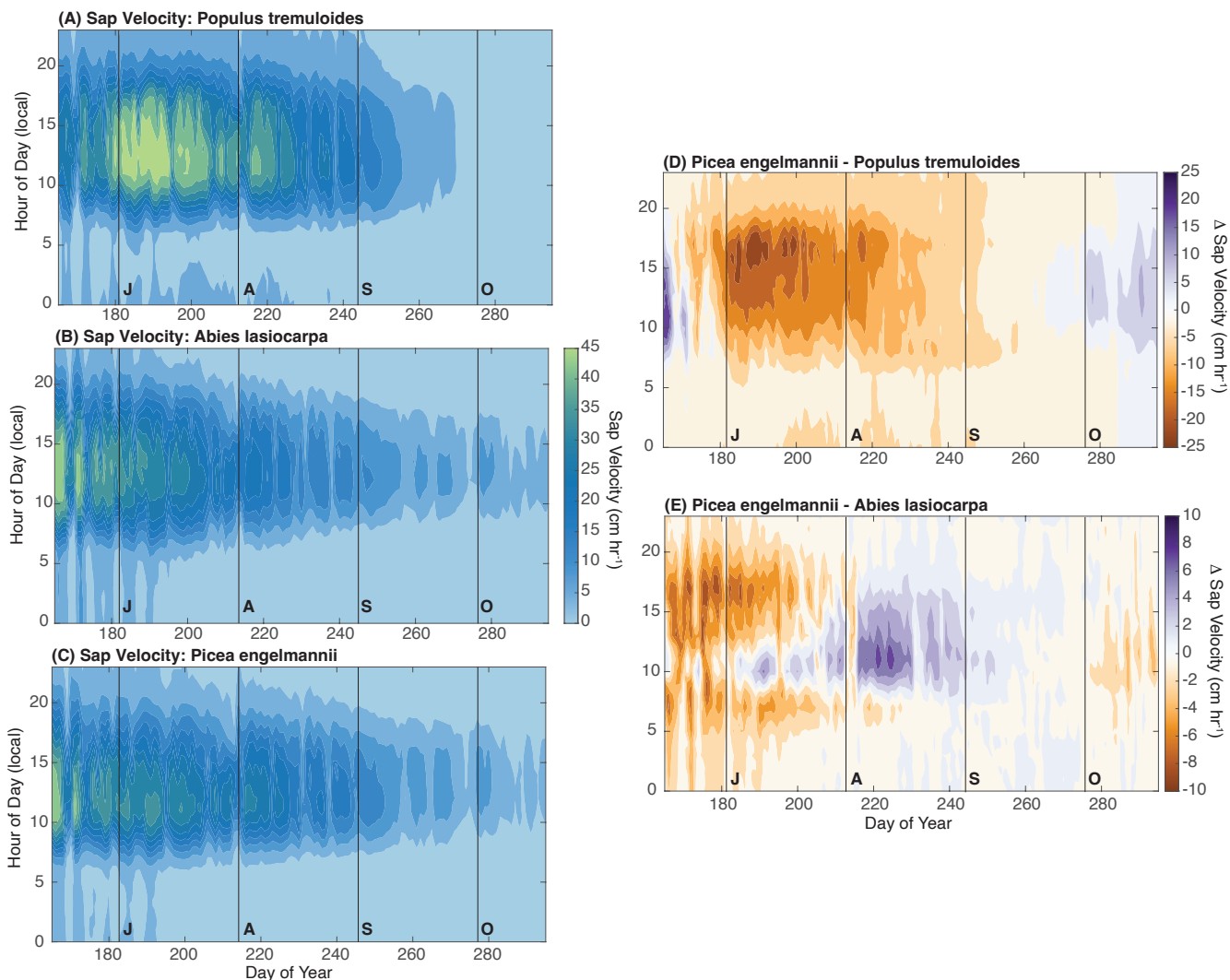

**Figure 2.** (A) Sap velocity averaged across all Trembling aspen trees between years as a function of Hour of Day (y-axis) and Day of Year (x-axis). (B) As in A but for subalpine fir. (C) As in A but for Engelmann spruce. (D) Differences in sap velocity between Engelmann spruce and Trembling aspen. (E) As in D but for Engelmann spruce and subalpine fir though note differences in scale bar to accentuate the more subtle differences between conifer species.

shorter period of activity, the peak rates and average sap velocity for aspen were substantially higher than both conifer species. However, we did not capture autumn through spring water use by conifers and the conifers account for ~60% of the hillslope sapwood area (Figure S3), so we estimate that aggregate hillslope water use was likely similar between the conifers and aspens.

The years during which this experiment were conducted (2019-2022) were characterized by both periods of higher than normal spring snowpack with lower than normal summer rains (2019) and lower than normal spring snowpacks with higher than normal summer rains (2021-2022) (Figure 3). We use May 1 snowpack as our spring snowpack indicator because this metric captures the magnitude of the snow reservoir present during the period when conifers begin to show high levels of activity and can therefore directly utilize inputs from melting snow. We note that over the last 40 years, there appears to be a slight negative relationship between May 1 and July-August (JA) precipitation as noted previously (Gutzler and Preston, 1997). This implies that trees are likely to generally experience contrasting inputs of winter vs. summer precipitation on typical years. Using this analysis on precipitation seasonality, 2019 ranks among the highest years for May 1 snow and among the lowest years for JA precipitation. In contrast, the period between 2020-2022 was characterized by below average May 1 snow and above average JA rainfall, illustrating the strong contrast in precipitation seasonality among the experimental seasons. While we did not include 2020 in our sap velocity analysis, we highlight this year in the figure to emphasize a sustained multi-year difference in precipitation seasonality over the last three years of the experiment relative to 2019.

Prior to Day of Year $\sim$200, almost all the trees (88%) showed an increase in sap velocity in 2019 relative to 2021/2022. This result illustrates how an extended spring snowpack sustains the trees water demands during the fore summer drought period (Figure 3b). However, by the end of the growing season less than half of the trees had retained higher total averaged sap velocity. In fact, averaged across all trees, there was a slight decline in cumulative sap velocity in 2019 relative to 2021/2022. This result illustrates two key findings: (1) a historically large spring snowpack like 2019 did not universally enhance water use and (2) across this population of trees, the buffering effect of the more active monsoon seasons of 2021 and 2022 appeared to fully offset the presumed negative effects of a lower snowpack. This result is consistent with results from Strange et al. (2023), who utilized a network of tree ring records across the southwestern US, to show that variations in monsoon rain were able to fully offset the deleterious effects of winter drought on tree stress.

## 3.2   Isotopic data

To place the patterns of sap velocity into a context of seasonal water source utilization, we rely on the twig water isotope samples. As expected, the twig water samples fall to right of the well-defined Local Meteoric Water Line - indicative of evaporative enrichment relative to precipitation inputs (Figure 4a). The isotopic ratio of the twig water measurements encompass a range of almost 20‰ in $\delta^{18}$O, showing the diverse spectrum of water sources across species and the hillslope over the three years of measurements. Based on the results from the mixing model, we estimate that - when weighted by sap velocity - the trees relied on $\sim$60% snowmelt though this value varied significantly across space and species. The hillslope receives $\sim$80% of its precipitation inputs in the form of snow so if the trees were relying on a mixture of seasonal water sources that purely reflected precipitation inputs, the reliance on snow would be higher than the $\sim$60% value we observed. In contrast, $\sim$90% of streamflow is supported by snowmelt (Carroll et al., 2020), which when combined with the twig water isotopes show that summer rains are preferentially routed through transpiration.

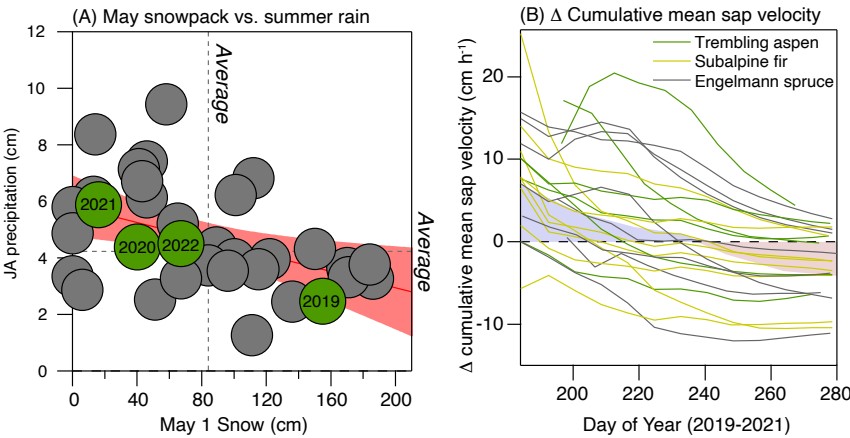

**Figure 3.** (A) The relationship between May 1 snowpack and July and August rainfall from the nearby Gothic Weather station (Faybishenko et al., 2023). The 4 years encompassing the study are labeled in green and a trend line with 95% confidence intervals along with the mean values for each seasonal precipitation inputs are indicated on the figure in pink. (B) The difference in cumulatively averaged sap velocity between 2019 and 2021/2022 for each tree that had continuous measurements during each of those three growing seasons. The shaded area is the average of all trees to indicate the mean sap velocity response between the contrasting precipitation years.

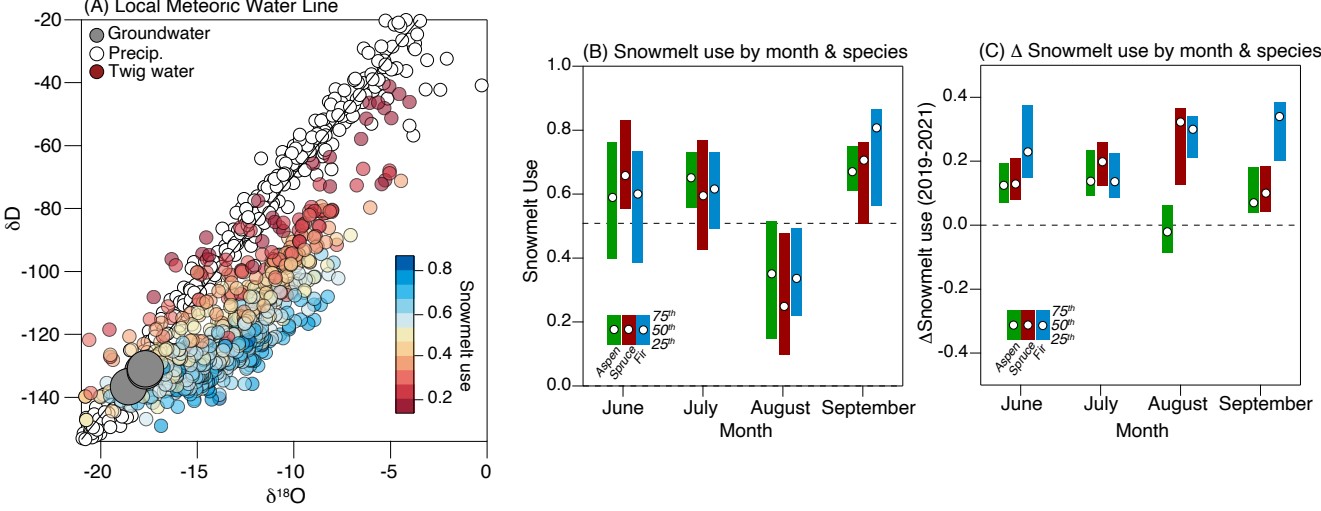

**Figure 4.** (A) Local meteoric water line for precipitation samples and the three years of collected twig and soil water samples. Samples are colored based on the inferred snowmelt reliance derived from the mixing model. (B) Monthly averaged use of snowmelt per species based on all available data. A value of 1.0 corresponds to complete dependence on snowmelt water. (C) The difference in reliance on snowmelt between 2019 and 2021.

The twig water isotope data shows that use of seasonal water sources varies through the growing season and between years. During June and July, snowmelt water accounted for 70%-80% of the water used while the importance of summer rain only prominently emerged in samples collected in August (Figure 4b). Interestingly, a return to use of snowmelt as a water source occurred in September indicating a reliance on groundwater or older snowmelt waters retained in the soil near the end of the growing season. Sap velocity is generally negligible during this period so the water source in September is not consequential to the water budget but may be critical for sustaining low levels of tree activity deeper into the fall after summer rain inputs have mostly been lost to evapotranspiration. As expected, we note a consistent increase in reliance on summer rain in 2021/2022 relative to 2019 across species and time (Figure 4c). The one notable exception to this pattern is the similar interannual use of summer rain in aspens during August. This may reflect the significantly lower interception rates for aspen (Thomas, 2016) which allowed utilization of the limited summer rain inputs in 2019 that did not penetrate the conifer canopy and/or the fact that extended spring snowpacks (as in 2019) can enhance summer rain use by allowing a higher density of surface roots to stay active during the fore summer drought (Bailey et al., 2023).

In order to assess partitioning of water sources between species, we filtered the isotopic data to look only at differences in water sources between co-located species sampled simultaneously. As shown in Figure 5, we can see that all three species utilized a statistically similar water source during June and July but then a distinction between aspen and conifers started to emerge in late July (∼day of year 200) and reached a maximum in species partitioning in early August. For comparison, there was never a statistically significant partitioning between Engelmann spruce and subalpine fir. The most obvious interpretation of the difference in water sources between aspen and conifers, emerges by comparing the soil moisture data from underneath adjacent aspen and conifer crowns (Figure 5d and S6). We see that the increasing reliance on summer rain by aspens as seen in the twig water isotopes corresponds to the periods when summer rains infiltrate under the aspen but not the conifer stands.

### 3.3 Integration of isotopic, sap velocity and stand structure data

By combining the estimates of tree snowmelt-reliance from stable isotopes and interannual variations in sap velocity (i.e. Figure 3b), we find that trees more reliant on snowmelt were those that benefited most from the large spring snowpack in 2019 (Figure 6a). Although this seems to be a self-evident result, it validates the idea that variations in the seasonal origin of a tree's water source influences its response to changing seasonal precipitation inputs. This is a link that has been inferred from previous studies but rarely proven explicitly. As can be seen in Figure 6a, the sensitivity of a tree to interannual variations in snow inputs was not, however, clearly predicted by species with examples emerging of individual aspen, fir and spruce responding in opposite directions to the precipitation differences between 2019 and 2021/2022. Instead, the results show that at two of the sites (#1 and #4) all of the trees were more active during 2019 and at the other three sites (#2, #3 and #5) the trees were less active during 2019 (Figure 6b). Site #1 is the lowest elevation site and is exclusively aspen while Site #4 is near the top of the hillslope and exclusively conifers (Figure 1). The species and elevational contrasts between these sites shows that the common response of the trees at these sites to variations in snowpack was not associated with a particular species trait nor the elevational

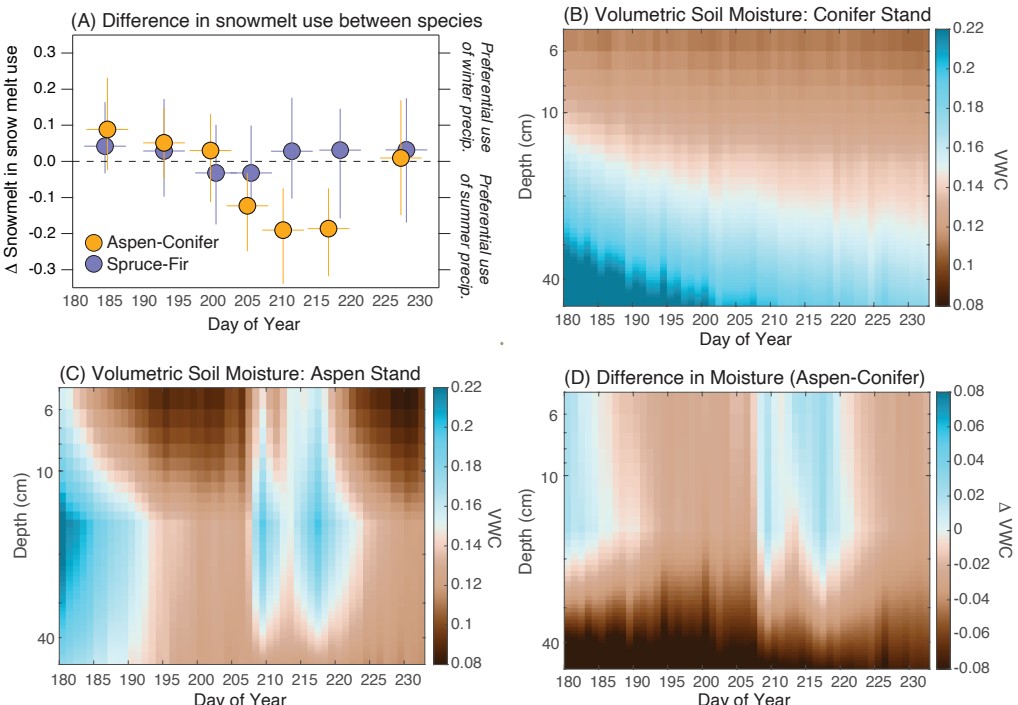

**Figure 5.** (A) Difference in reliance on snowmelt between Trembling aspen and both conifer species (yellow), and the difference between the two conifer species (purple). The data were generated by only comparing snowmelt use between co-located and simultaneously sampled twig water samples. The data were binned into 10-day windows to capture dynamics that were occurring at the sub-monthly scale. Error bars capture the 25th and 75th percentiles around each of those 10 day bins. (B) and (C) Volumetric water content under adjacent aspen and conifer stands interpolated from measurements at 5, 15, and 50 cm. The site location is indicated on Figure 1 and the data are available from (Simonpietri and Carbone, 2023). (D) Difference in volumetric water between aspen and conifer stands illustrating the moistening of surface soils under the aspen stands in late July and August.

position on the hillslope. The most conspicuous characteristic shared between these two sites is that they have similar sapwood to ground area values ($\sim$38 cm$^2$ m$^{-2}$) that are approximately three times greater than the other three sites ($\sim$12 cm$^2$ m$^{-2}$) and

well above the typical values for forested areas on this hillslope (Figure S5).

Because this was not a controlled thinning experiment where we have incremental changes in density across stands of a single species, it is difficult to interpret the significance and causal nature of this pattern. We can, however, show through a simple probabilistic argument that the likelihood that all the trees in the dense stands behaved similarly and in opposition to

the behavior in the more open sites was not random. There was a total of 26 trees instrumented across these stands that had continuous sap velocity measurements during 2019, 2021 and 2022 and 10 of these trees showed a positive response to 2019. All six of the trees within the two dense stands were within the population of 10 trees that showed an increase in sap velocity

during 2019. The probability that all 6 trees in the two dense stands would show the same response just through random sorting is 0.002 based on a Monte Carlo simulation (n=10000). Although this result does not causally link stand density to the way sap velocity responds to spring snowpack, it shows that there was non-random organization such that the different response between Sites #1 and #4 vs. Sites #2, #3 and #5 captured distinct ecohydrological behavior between these sites. As discussed above, differences in transpiration rate and seasonal water sources between stands of contrasting densities has been established in previous studies and this is thus not a surprising result (e.g. Tague et al. (2019)).

Although we have focused on sap velocity as a metric to understand tree responses to precipitation seasonality, we also generated stand-level transpiration estimates by scaling up these measurements using LiDAR tree crown data as described in Section 2.4. The peak estimated transpiration rates from these scaling estimates were comparable to previously published estimates of evapotranspiration from a flux tower in a riparian zone at the base of this hillslope (Ryken, 2021; Ryken et al., 2022), providing validation for this approach. The average transpiration rates across sites were nearly linear related to sapwood area - as expected and noted in previous studies (Berry et al., 2018) (Figure S4). This translates to a factor of 3-4 difference in transpiration rates between the open vs. closed stands. The transpiration data provides a distinct perspective on the ecohydrological response to the large spring snowpack of 2019. For example, Sites #1 and #4 showed a $\sim$5 cm h$^{-1}$ increase in sap velocity in 2019 whereas Sites #2, #3 and #5 showed a decrease of about the same magnitude. This would suggest a response to 2019 between the dense and open stands that was symmetrical but opposing. However, after scaling this response to stand-level transpiration, it translates to a $\sim$4.3 kg H$_2$O m$^{-2}$ h$^{-1}$ transpiration increase at the dense stands vs. just a $\sim$0.5 kg H$_2$O m$^{-2}$ h$^{-1}$ decrease in the open stands. This indicates that because the denser stands have a higher peak capacity to move water their potential variance is much larger even if the individual tree level responses are similar across stands.

While emphasis thus far has primarily been on drivers of interannual sap velocity variability, we also used the merged isotopic and transpiration data to shed light on whether there were optimal seasonal water utilization patterns over the period of this experiment. To assess this, we compared peak transpiration rates for individual trees - as a proxy for the hydraulic capacity of a tree - against the water source used by the tree (Figure 7). We see from this analysis that use of a diverse mixture of water sources characterized by neither a strongly snow- nor rain-dominated mixture led to the highest peak transpiration rates (Figure 7). Those trees with particularly high reliance on either snow or summer rain use (i.e. a more narrow range of water sources) tended to have lower peak transpiration rates than those trees which have mixed water sources albeit with a preference for snow of $\sim$70%. These patterns can be mapped onto the hillslope context as portrayed by the Topographic Position Index (TPI) to show that sites with lower to moderate TPI values (low slope, convergence zones) exhibited more diverse mixture of water sources and higher individual transpiration capacity than those sites with higher TPIs (i.e. in hilltop areas). The relationship between TPI and changes in the dominant water sources used by trees was also observed in a Douglas fir dominant hillslope in the northern Rocky Mountains (Hoylman et al., 2018).

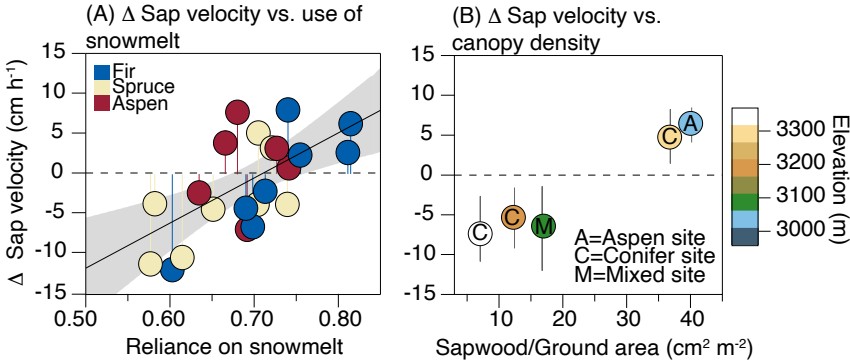

**Figure 6.** (A) The relationship between differences in averaged sap velocity between 2019 and 2021/2022 vs. the weighted average reliance of use of snowmelt for each tree. The positive relationship implies those trees that rely more heavily on snow are those that were more active during the large spring snowpack year of 2019. (B) The differences in sap velocity between 2019 and 2021/2022 organized by the stand density that each tree fell within (Figure 1). The error bars capture the range of Δ sap velocity for all trees within the stand. Note that only 5 stands are present in this figure because Stand #6 did not have sufficient continuous data to generate a difference estimate between years.

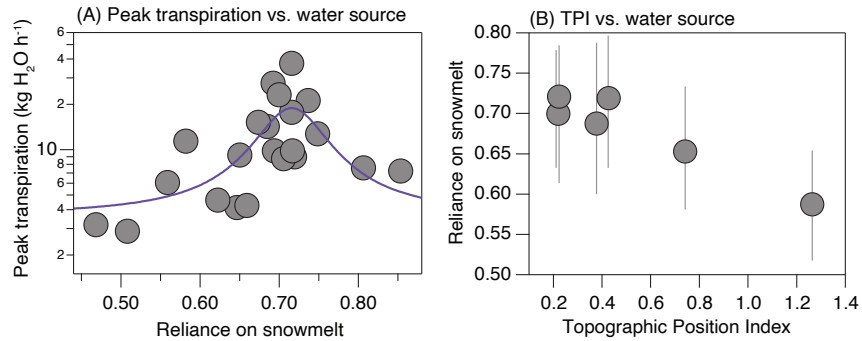

**Figure 7.** (A) Relationship between peak transpiration reached through the experimental period and the average snowmelt reliance of the tree. Note the units here are kg $H_2O$ $h^{-1}$ derived by multiplying sap velocity by sapwood area. A 3rd order polynomial fit was added to the figure to highlight the structure and location of the optimum value. (B) Relationship between snowmelt reliance and topographic position index (TPI) of each site with the range of snowmelt reliance captured by the error bars. This figure has six sites while the previous figure only has five because while we did not have sufficient data from Site 6 to assess differences between years, we did have data to estimates of peak transpiration.

## 4 Discussion and Conclusions

There is extensive literature based primarily on isotopic data and modeling that shows the importance of snowmelt as a water source to support the transpiration demands of common tree species across the subalpine forests of the western US. As spring

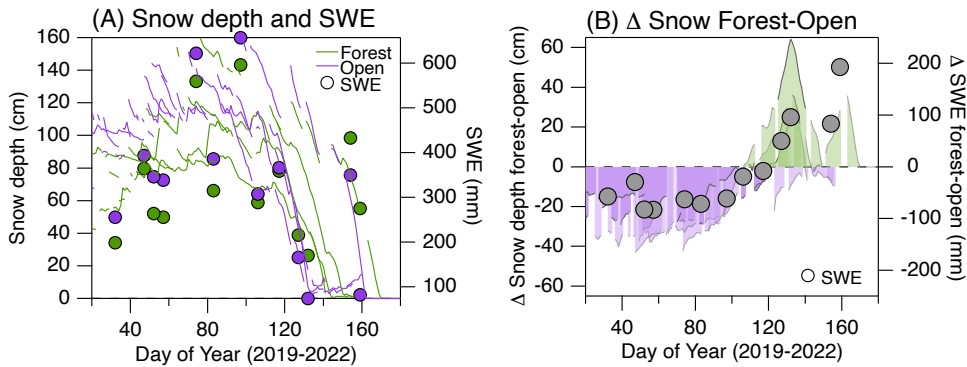

**Figure 8.** A) Seasonal trends in snow depth shown as lines (left axis) and SWE shown as dots (right axis) derived from continuous snow depth sensors and periodic snow pit measurements for adjacent open and forested sites as indicted on the map in Figure 1. The multiple lines per site capture data from 2019, 2020 and 2021. These data were published by Bonner et al. (2022). (B) The average difference in snow depth and SWE between the forest and open sites for 2019-2021. The results indicate that while the forest site had generally lower snow during the winter starting from day of year 120 to day 160, there was a persistently larger snow pack in the forested site.

snowpack declines and growing season evaporative demands increase, it remains unclear the extent to which these changes will
limit forest productivity, influence downstream water subsidies and increase susceptibility to disturbance. Here, we utilized a distributed network of sap velocity sensors and stem water isotopes to generate quantitative information on the magnitude and sources of water-use at the species- and stand-scale for a hillslope in an extensively studied watershed in the Upper Colorado River Basin. To a first order, the sap velocity data mirrored results from a similar study by Pataki et al. (2000) done in this region over 2 decades earlier. Notably, aspens had significantly higher instantaneous sap velocity rates and were more responsive
to small summer rain inputs than the subalpine fir or Engelmann spruce trees. The aspen were able to sustain high levels of activity through the summer months even during a growing season like 2019 that experienced a sustained fore summer drought period and weak monsoon (Anderegg et al., 2013; Sloat et al., 2015). On the other hand, many of the individual conifers during 2019 began to show significant declines in water-use by early July and sustained only minimal flow rates by August even when small rainfall inputs emerged. These differences reflect the higher threshold for embolism and reduced rainfall interception by
aspen. We also document measurable differences in the seasonality and diurnal cycles between subalpine fir and Engelmann spruce such that the latter were more active later in the season and had reduced sap velocities during morning and evenings. This was consistent with some previous work documenting a higher sensitivity to vapor pressure deficit and lower sensitivity to soil moisture for Engelmann spruce relative to other common coniferous species (Oogathoo et al., 2020; Pataki et al., 2000). These differences are small relative to those between either conifer species and aspen and will be further analyzed in forthcom-
ing work that includes hydraulic trait measurements of these trees along with process-based modeling.

In our experimental design, we distributed the isotopic sampling and sap velocity sensors across stands that covered a range of conditions typical for this hillslope. The design included example stands of conifers, aspens and mixed aspen and conifers

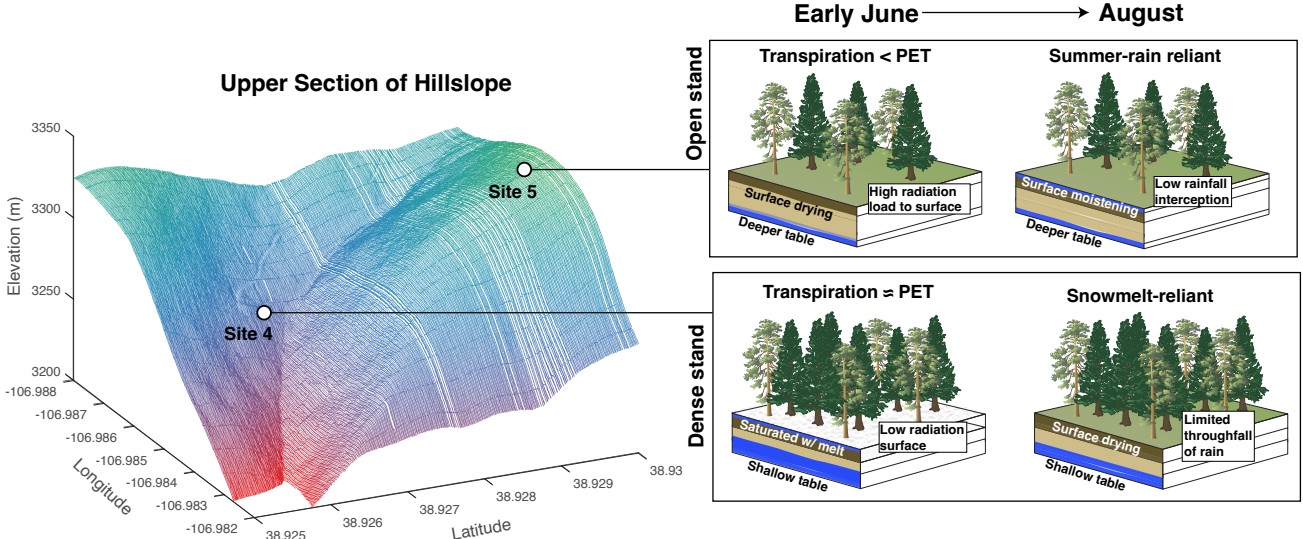

**Figure 9.** Schematic view showing the way hillslope position and canopy structure can influence seasonal water source access and stand density.

that had both sapwood to ground area densities that were typical for the hillslope as well as stands with densities near the maximum of the hillslope. We observed differences in stand-level transpiration across these stands that were linearly related with density and varied in their transpiration rates across a factor of five ($\sim$2.0 vs. $\sim$12.0 kg $H_2O$ m$^{-2}$ h$^{-1}$) (Figure S4). The linear response of transpiration to stand density has been widely noted in other sap velocity studies from a variety of forests including similar sub alpine systems (Tor-Ngern et al., 2017). In an earlier study that utilized one season of data from this same sap velocity network, Ryken (2021) found that two of the densest sites - corresponding to Sites #1 and #4 in Figure 1 - displayed transpiration rates that were comparable and/or exceeded June and July latent heat flux rates from an eddy covariance in the riparian zone at the foot of this hillslope (Ryken et al., 2022). In contrast, the open stands always had transpiration values well-below the observed evapotranspiration (ET) rates derived from eddy covariance. This result suggests that the densest sites in the network were operating in a state during June-July when transpiration accounted for virtually all the ET and was similar to estimated potential ET (PET). Transpiration was thus fully energy-limited at least during the early periods of the summer prior to the dry down of soil moisture in late July (Fig. 9). These transpiration rates are likely the product of a long term acclimation at a few localized areas of the hillslope where persistently higher amounts of soil moisture or shallow groundwater to support an optimum level of sapwood area. Unfortunately, we did not have a network design to test the incremental effect of stand density on the stand density when transpiration accounted for all of ET but it is clear that the open stands had transpiration values that always fell well below ET. Unlike, results from some previous studies such as Bréda et al. (1995), we did not see lower rates of transpiration from individual trees in the dense stands relative to the open stands indicating that even with the higher competition for water there was virtually no competitive limitation on water for weeks after snow had completely

melted out. The dense sites also displayed an order of magnitude larger range in interannual transpiration variability (Figure S4) indicating these uncommon areas of the hillslope are likely important drivers in temporal variations in how much snowmelt is routed to transpiration (Faramarzi et al., 2009).

To develop a more mechanistic perspective on how species- and stand-level sap velocity related to access of specific water sources, we utilized the results from the stable isotopic mixing model. One of the most conspicuous results that emerged from the isotopic data is the emergence of partitioning between co-located aspen and conifers during the later portions of the growing season (Fig. 5a). There was a clearly higher reliance on summer rain inputs by aspen relative to conifers as the growing season
progressed and this was apparent in multiple years and multiple stands and confirms results presented in earlier studies from other sites in the region (Anderegg et al., 2013). The reason for the species-level partitioning can be explained by multiple co-existing processes. Conifers have 3-4 time higher leaf area per basal area than aspen (Pataki et al., 2000), which results in up to 40% higher interception by conifers (Thomas, 2016). Soil moisture profiles from this hillslope and a nearby site in the watershed, as reported by Carbone et al. (2023) (Figure S6), confirm that soils at 5 and 15 cm depths were often non-responsive
to small summer rain events under conifer stands. This result shows how the partitioning of water sources between species does not, per se, require any explanation relating to belowground competition for a common water pool (Berkelhammer et al., 2022) but rather results from differences in aboveground traits relating to leaf area and interception. LaMalfa and Ryle (2008) also noted differences in the permeability of soils beneath conifers and aspens which could further increase how much of the summer rainfall infiltrates below aspen canopies. The combination of lower interception, increased soil infiltration and higher threshold
for embolism all collectively increase the ability of aspen to rely on summer rain inputs. An easily overlooked aspect of the differential water sources between conifers and aspens is that aspens regularly exhibited sap velocity rates that were twice that of conifers (Figure 2) while exhibiting a 20% higher relative reliance on summer rain (Figure 5a) which implies the conifers and aspens were actually transpiring similar amounts of snowmelt water but the aspens were also mixing summer rain into the transpiration stream. All the species thus exhibited a similar reliance on one water pool (snowmelt water) while aspen were
able to also take advantage of the second pool associated with periodic summer rains. As illustrated by Figure 7, the trees that utilized a mixture of summer and winter precipitation (primarily aspen) yielded the highest rates of instantaneous transpiration.

The spatial patterns and species dynamics described above illustrate key properties of forest water use in this system but the interannual changes are needed to understand the sensitivity of these systems to a shift in snow hydrology. The interannual
comparisons show that snowmelt-reliant trees were more active in 2019 and that about 40% of the differences in individual tree sap velocity rates between the contrasting 2019 and 2021/2022 growing seasons can be explained by the seasonal origins of the water used by the trees. Previous studies have drawn inferences about reliance on snowmelt via relationships between satellite greenness metrics and snowpack inputs in other subalpine system (Berkelhammer et al., 2017; Trujillo et al., 2012) but we now make an explicit connection between snow utilization and tree activity. As described in studies such as Martin et al. (2018)
and Kerhoulas et al. (2013), there are measurable differences in reliance on seasonal water sources within a hillslope owing to factors such as hillslope position and stand density. Indeed, we see evidence of distinct water use patterns and responses to

the 2019 vs. 2021/2022 growing seasons between the five stands. The dense aspen and conifer stands both were more active during 2019 and the open stands that included both mixed and exclusively conifer stands were more active during 2021/2022. As reviewed by Tague et al. (2019), there are myriad hydrological processes affected by stand density that influence forest utilization of different seasonal water sources including interception of snow and rain, changes in surface radiation loading, tree rooting depths, soil permeability, competition among trees and with understory plants. In contrast to the results we see here, both Kerhoulas et al. (2013) and Sohn et al. (2014) noted that in denser (unthinned) stands, trees were more snow reliant. They argue that this was the consequence of increased canopy interception of snow, a decrease in shallow-rooted understory plants and shallower rooted trees in the unthinned stands. Among the sites instrumented here, there was minimal understory and no evidence of taller trees in the thinner stands (Figure 1). The absence of shallow-rooted understory plants here would likely accentuate the way trees in the open stands were able to benefit from the increased throughfall of summer rain. Furthermore, while snowpack reached higher peak values in open areas, the snowpack persisted later into the growing season in forested stands allowing surface soils to be consistently recharged during the early period of the growing season (Figure 8). The relationship between SWE and canopy density is non-linear and varies with background climate (Dickerson-Lange et al., 2021; Lundquist et al., 2013) but the limited results from this hillslope suggest that snowpack persistence might be one of the mechanisms that would be favorable to increasing access to snow melt waters in the dense stands (Figure 9).

In considering how the results here contrast from previous work showing lower snow reliance and more tree-level water stress in dense stands (e.g. Belmonte et al. (2022)), it is important to note that many comparable studies took advantage of controlled thinning experiments within single species stands. With the design of this network we cannot, for example, conclude that the similar response to the large spring snowpack of 2019 between the dense aspen stand (Site #1) and the dense conifer stand (Site #4) were the result of a common mechanism. The conifer stand may have benefited from the impact of stand density in increasing the persistence of snowpack while the aspen stand may have been able to take advantage of the large snowpack to sustain active shallow roots longer through the drought period as described by Bailey et al. (2023). It is also notable that thinning experiments like those utilized in studies such as Sohn et al. (2014) or O'Donnell et al. (2021) were intentionally designed to understand how changes to aboveground properties influence ecohydrology with the explicit goal of decreasing tree-level water stress. The experimental design of these studies would intentionally try to minimize differences in soil properties or groundwater depth between stands that would naturally lead to differences in canopy structure. On the other hand, the dense stands where we made measurements likely reflect topographically-mediated convergence and/or the presence of deeper soils that support stand structures with high potential for water use and persistent connections with water recharged by snowmelt. So while the dynamics we observed here may appear to contrast results from thinning experiments they actually provide potentially complementary information on how above and belowground properties influence the sensitivity of a forest stand to changing snow inputs. The above and belowground effects may enhance the differential responses between stands such as an instance where access to shallow ground water supports higher canopy density while simultaneously leading to a snowpack that lasts longer into the growing season. We also note potentially similar dynamics in an unmanaged stand in the the Sierra Nevada as noted by Goodwin et al. (2023) who documented an inverse relationship between the isotopic ratio of tree

cellulose and stand density. Though further work is needed to separate how much of the isotopic signal in the cellulose reflects changes in the seasonal origins of source water vs. the impact of open stands on crown conductance or surface evaporation.

Although there are limits to the generalizable conclusions that can be drawn from this study, we use the results to pose some hypotheses about the responses of subalpine forests in the western US to changes in snowpack. In unthinned forests, the largest responses to changes in snowpack in terms of changes in transpiration will be centered on those locally dense stands that have developed in the context of high reliance on snowmelt. The effects would likely be more pronounced in conifer stands that have a generally lower capacity to utilize summer rain inputs. While these sites have historically had higher access to soil moisture

that gave rise to the higher leaf areas, they are more likely to be vulnerable to dieback and thinning in the likelihood of a low snow future due to the combination of their locally high water demands coupled with their higher rainfall interception rates that minimize access to summer rain. In a trajectory where the dense stands experience dieback, this natural thinning process could lead to the emergence of more shallow rooted understory and potentially deeper root system for the trees, thus maintaining high reliance on snow but an overall reduction in transpiration due to the loss of stem and leaf area. The consequences of this for

groundwater recharge and streamflow generation are difficult to predict. For example, how will the rate that spring snowpack is being reduced compare to the structural responses of the forests? How will these structural changes impact buildup of winter snow vs. timing of spring melt? Will shifts in plant phenology and increased reliance on summer rain offset or buffer these forest responses?

We recognize a number of important limitations of this work that future field and modeling experiments could address. Firstly, we discuss the differences between years (2019-2022) only with respect to changing precipitation seasonality. This is because the differences in snow inputs were the most notable difference between years but use of a more process-based modeling approach would allow us to quantitatively interpret the differences in sap velocity not only to changing snowmelt inputs but also to interannual differences in the timing of rain events, VPD and temperature. Also, the design of our network was not

intentionally focused on canopy structure and thus we do not have data along a continuum between very dense or open sites. It was thus not possible to infer how stands that fell between the more open and closed end members behaved - which would be needed to scale up towards and aggregated hillslope estimate of transpiration changes to snow input variability. Controlled thinning in experimental plots that include different species composition and different initial densities would be the optimal testbed to explore the continuum of these effects. Furthermore, adding measurements of tree hydraulic properties to link their

responses to changes in water access will help to develop an understanding of the variance in behavior that was not driven just by changing snow inputs. In particular, we see some evidence that individual trees with access to diverse water sources can reach the highest peak transpiration rates but the mechanisms for this relationship (e.g. root profiles or root hydraulics) remain unexplored in this study. Despite these limitations, our work links key observations about forest dynamics and changing hydrology that could help guide forest management decisions in such a way to optimize the ecological utilization of summer rain

in regions like the southwestern US that receive regular summer rain inputs. Furthermore, the work provides needed bench-

marking information for future simulations of coupled ecological and hydrological processes at the watershed scale.

*Data availability.* The data associated with this manuscript is available here: https://data.ess-dive.lbl.gov/view/doi:10.15485/1647654 (doi 10.15485/1647654)

*Author contributions.* MB wrote manuscript, led analysis of data, assisted with field deployment and design and resource acquisition; GFP edited and wrote manuscript, led field design and data acquisition; CS assisted with editing manuscript and resource acquisition; FZ, WT, LH, JB, KI, AK, MC, KF, WB, MW, MSC, IB, AR, RM, DG, MR, ES and KHW all provided critical data sets for the analysis.

*Competing interests.* The authors declare that no competing interests are present.

*Acknowledgements.* We would like to acknowledge all the supporting staff at the Rocky Mountain Biological Lab. MB and CS acknowledge
Department of Energy Office of Science, Biological and Environmental Research, Award Number: DE-SC0019210. The work of WT, AK and KI was supported by the US Department of Defense / National Defense Education Program through the Educational and Research Training Collaborative at the University of Illinois Chicago, Grant HQ-00342010037. MSC acknowledges funding from DOE BER (Grant no. DE-SC0021139 and DE-SC0024218) and a RMBL graduate fellowship to AS. RWHC acknowledges US Department of Energy Office of Science under the contract DE-AC02-05CH11231. This material is based upon work supported by the National Science Foundation under Grant No.
1761441 to MR and ES. This material is based upon work supported as part of the Watershed Function Scientific Focus Area funded by the U.S. Department of Energy, Office of Science, Office of Biological and Environmental Research under Contract No. DE-AC02-05CH11231.

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
