# Peer review of "Canopy structure modulates the sensitivity of subalpine forest stands to interannual snowpack and precipitation variability"

_EGUsphere, 2023_

## Author Comment (AC1)

**Response to Reviewer #1**
Thank you to Reviewer #1 for the constructive feedback on the manuscript. Detailed responses are listed below.

*Here are some possible overlooked aspects; the paper might benefit from addressing the long-term climate change implications on these dynamics in more detail, such as the potential for increased frequency of droughts or low snowpack years and its impact on forest ecosystems. Also, some crucial figures in this version seem to be a bit too small, making it difficult to read without zooming a lot.*

We will add additional details about how these results inform us about the effects of future shifts in drought and/or low snowpack years. There was a mistake in the figure size designation in Latex that led to the small figures. We will adjust the figure and axes text sizing.

*Comments by line.*

*Line 35, incomplete citation. (and more citations also inclmplete on the ms.)*

There were some errors in the bibtex file that will be rectified to fix, for example, the "Strange et al.," reference.

*The hypothesis in line 90 is not fully clear to me. Does it mean that the nature of water availability will determine the stand sensitivity to changes in annual precipitation distribution? The second hypothesis made me think about the fact that canopy structure is a much more complex and dynamic outcome that depends on site conditions and not just water availability. Is having access to snowmelt indicative of deeper and better soil conditions?*

Similar concerns were raised by Reviewer #2 about the hypotheses being stated in a confusing way. Below is an example of a revised version of the text that clarifies the summary statement for the manuscript.

*We show that: (1) trees with higher reliance on snowmelt water are more sensitive to interannual changes in snowpack; and (2) that trees in denser canopies were more reliant on snowmelt water. From these observations we hypothesize the presence of a positive feedback where canopy density reflects long term access to snowmelt. The denser canopy, in turn, further sustains reliance on snowmelt by modifying shortwave absorption at the surface and rainfall interception.*

*Line 130 related to the deuterium correction might be a good idea to compare notes with https://hess.copernicus.org/articles/26/5835/2022/ that seem to relate this methodological "offset" with the volume of the water samples exposed to the CVD.*

We will include this reference and a few others while also reporting the percent recovered water from our samples.

*In methods, we have the seasonal origin index, which has not been introduced or has been introduced well enough.*

More details on the SOI will be presented earlier and with more detail.

*Figure 2: I recommend using one more color on the gradient of A, b, and C  so the upper sap velocities >40 are easily distinguished between trees. Also, I would add the months to the figures. Not everyone is familiar with relating DOY and the months…*

We will utilize a multi-color bar and add months adjacent to DOY.

*Figure 4 could be larger for easy reading, and the dots of stem water might be better to plot with smaller dots and maybe somewhat transparent. (Just a suggestion). Also, what is the difference between SOI and the relative or reliance snowmelt use? Maybe it would be nice to make this clearer, as I am getting confused  with the rest of the figures*

This figure was accidentally made too small by a coding error in Latex. We can separate the stem (twig), soil and precipitation waters to make this more clear.  We rely primarily on the "snow reliance" metric based on the mixing model not the SOI and, frankly, it may be valuable to just remove SOI as it is not integral to the paper and is somewhat redundant relative to the results from the mixing model.

*Line 250: I am not sure if I am following why there is reliance on seasonal rain if the snowmelt represents 80% of the annual.*

This is a good question and requires some additional discussion.  Rain that falls during the growing season moistens the surface soils during periods of active transpiration. This is especially important because the intensity of summer rain tends to increase in July following an extended dry period in June. The delivery of this moisture is thus crucial to avoid extremely low matric potentials in the surface soils. On the other hand, a significant amount of snowmelt is routed to runoff and groundwater recharge.  So, while the hydrological inputs to the watershed are dominated by snow, the water used by transpiration is more evenly distributed between snow and summer rain.

*Line 365, maybe there is a bit of an offset between this and the message from the introduction. Maybe it would be good for the readers to be able to connect more clearly with this reminder on the discussion, and the canopy structure is supposed to be a key element of the manuscript, which was a bit in the shadows, in my opinion, throughout the manuscript also edit the citations years.*

We appreciate this comment and I think it reflects similar concerns regarding some confusing structure in the framing of the manuscript.  In the revision process we plan to do a better job of separating what are the core identifiable conclusions to be drawn directly from the observations from hypothesized processes and mechanisms driving these relationships.

*Line 385 might be something to consider: reference the result sections to some figures or particular sections where this is supported instead of sending the readers to the supplementary info. It seems like this should be an important part of the manuscript.*

We can bring the SWE results into main body of the manuscript, present it with the Results and reference accordingly.

*On the supplemental material,*

*I would recommend adding a list of the figures in the first page.*

We can add a Table of Contents for the Supplement.

*FigS2 seems to be cut on the text above the figure also could be nice to make reference to the literature and the methods on how this was addressed.*

We will address the cropped image and add additional text to explain how we determined the scale of the deuterium offset.

*FigS3 here is something related to the SOI and relative contribution that was creating some confusion explained above, where it was not clear to me why the SOI is used in the main ms.*

Yes, we can either bring this into the main body or, alternatively, remove the SOI analysis as it is not needed for the analysis.

*FigS6. I'm not sure if the y-axis labels are correct in both figures.*

The units can be adjusted to mm h-1, reported as "transpiration" and also changed so Panel A is mm h-1 and Panel B is difference in mm h-1 between years (i.e., presented as a "transpiration anomaly").

**Response to Reviewer #2**
We would like to thank Reviewer #2 for the detailed response and careful consideration of this manuscript. While we obviously see more merit to this work than the reviewer did, we acknowledge the important concerns they raised regarding the general inferences on ecohydrological processes we tried to draw from a relatively small observational dataset. With better clarity on separating robust conclusions derived directly from sapflow and isotope data from conjecture on mechanisms, we think the concerns raised can be largely addressed. We respond to the individual reviewer concerns below.

*1) The overall narrative and conclusions go beyond what should be inferred from the data. While the authors admit that the investigation was not established to pursue the narrative as it is presented, the truer statement is that the investigation falls short of what is needed to support the arguments regarding the role of canopy structure (which is reinforced by the scarcity of statistical tests).*

We appreciate the critical feedback raised by the reviewer, and we argue that some simple statistical analysis can support the idea that canopy structure is an important determinant of how trees respond to changing snowpack on this hillslope. Two of the stands we instrumented had similarly dense canopies, while the three other stands had similarly open stands. There was a total of 26 trees instrumented across these stands that had continuous sapflow measurements during 2019, 2021 and 2022. Six of those trees were within the two dense stands, and *all* of those trees showed an increase in transpiration during the large snowpack year (2019). There were 4 additional trees distributed across the three open stands that also showed a positive response. The probability that all 6 trees in the two dense stands would show the same response just through random sorting is 0.002 based on a Monte Carlo simulation (we will include a similar analysis in the revision). While this result does not prove that canopy structure *caused* the particular transpiration response to snowpack, it shows the response to changing snowpack was not random across the stands. Secondly, the trees that responded positively to higher snowpack included all three species, including functionally distinct aspen, showing that the response is not specific to the traits or water-use strategy of one species. This analysis can be fleshed out in a revision but illustrates a probabilistic argument to support the finding that canopy structure was an important determinant of how trees responded to changing snowpack.

*Far more study sites with varying ecosystem structure would be needed to support the argument that topography shapes density, or that density relates to sensitivity, let alone to assert "the progression towards a low snowpack future will manifest at the sub-hillslope scale in dense stands". The story-telling narrative style involves of web of speculations, making it challenging to connect evidence with statements made throughout. A reasonable level of evidence is crucially unmet in many areas.*

As noted above, we acknowledge that landscape-scale projections from this limited dataset need to be treated with more caution.  However, we'd also point out that Reviewer #1 came to an opposite conclusion i.e. they requested additional information about how these results could inform our understanding of how montane forests will respond to climate change.  So, while we recognize the need to be cautious in not over extrapolating from the data, we also recognize there is value to discuss the potential broader significance of these results.

*One key result is presented in line 281-283, "we see that the response of trees to the large spring snowpack of 2019 can best be predicted based on whether a given tree was located in one of the dense stands (i.e. where sapwood to ground area was ~38 cm2 m2) or open stands (i.e. sapwood to ground area ~12 m2 m2) (Figure 6)"; let's ignore the typo (it's not "m2 m2" but "cm^2 m^-2" ). Figure 6b shows 5 data points, and there is no aspen stand with low density and there is no mixed stand with high density, and thus there is no variation in density for a given composition of stand. This alone invalidates the primary argument of the paper (especially because we can see that species does matter in Figure 6a, Figure 4, and Figure 2). This critically small sample size does not allow for assessing stand effects.*

As discussed above, we can: (1) apply a simple statistical analysis to show that the common increase in transpiration in trees from the two dense canopy stands would not emerge randomly; and (2) we see aspen, fir, and spruce trees responding differently whether they are in open vs. closed stands.  There are aspects of these results that would benefit from more attention in the revision such as the role of tree to tree competition. This study is clearly not a full factorial analysis and, as noted in the manuscript, testing these ideas in controlled plots would be needed to prove this but the data suggests that species respond differently depending on the canopy density.  Nonetheless, we recognize the over-confidence in the way we presented this and our language muddied the distinction between observations and hypothesized mechanisms.

*2) There are major communications issues throughout, including use of imprecise language, omitted details, and insufficient copy editing.*

*Throughout, there is much suggestive and imprecise language, making it challenging to interpret and certainly not reproducible. Let us start with a few examples of uninterpretable statements:*

*-"we consolidate the numerous processes acting across sites by collapsing the measurement clusters…" (page 4 line 84)*

*-"it is also common to see areas where winter precipitation and groundwater may be of equal or greater importance to the species' water demands illustrating an overriding impact of hillslope context relative to species-specific traits" (line 64-65)*

*-"The sensitivity of annual tree activity to snow inputs did not clearly map onto species or elevational position on the hillslope." (line 280)*

*I do not know what these statements mean.*

We appreciate the reviewer pointing out examples of confusing language. As part of the revision process, we plan to clarify these sentences and choose simpler and more precise language. The manuscript went through several rounds of internal revision by the many co-authors. Consequently, we felt like the message and language choices were reasonable, but stepping back from the manuscript with a more critical eye on wording clarity is needed.

*The imprecise language is even evident in the (arguably) most important sentences of the paper, with these two hypotheses: "Within this context, we first test two fundamental hypotheses: (1) the seasonal origin of water used by trees influences a stand's sensitivity to changes in precipitation seasonality and (2) that canopy structure is a reflection of the source of water accessible to the trees in that stand." I cannot understand how hypotheses of "influences" and "is a reflection" could be tested, establishing murkiness early on.*

It is a shame that the hypotheses statements came across as muddled as noted by both Reviewers. This is a case where a single sentence was written and rewritten so many times its meaning got lost. We provided an example of a rewritten version of this sentence above in response to Reviewer #1.

*We show that: (1) trees with higher reliance on snowmelt water are more sensitive to interannual changes in snowpack; and (2) that trees in denser canopies were more reliant on snowmelt water. From these observations we hypothesize the presence of a positive feedback where canopy density reflects long term access to snowmelt. The denser canopy, in turn, further sustains reliance on snowmelt by modifying shortwave absorption at the surface and rainfall interception.*

*It is not just a writing-style issue because there are also major omissions and indicators of insufficient proofreading. Grammar problems occur throughout (e.g., a lack of appropriate punctuation).*

As stated, the manuscript went through many rounds of internal revision so I am sorry to hear that the reviewer found so many grammar problems - though specific references to these grammar problems would have been useful for us during the revision. We will continue to work on this during the revision process.

*Methodological details are too vague (although the Methods section is otherwise nicely structured). The sap flow study design is insufficiently communicated as it is unclear which trees and which trees in which sites were actually instrumented (Figure 1 only partially helps because there are places where the yellow circle could be over a deciduous or needleleaf tree, and there are issues such as only two yellow circles being visible in a site that supposedly had 4 to 6 trees instrumented).*

We appreciate the positive feedback on the Methods section structure. For the sap flow approach we can add additional details. The confusion the reviewer notes with Figure 1 is that we instrumented *2 trees per species per site* so for sites that were just aspen this meant just two

trees were instrumented.  For sites that had all 3 species, this meant 6 trees.  This was not communicated properly in the Methods text.

*It repeatedly says that 'live stems' were sampled for isotope data, which seems unfathomable for mature trees (were they live twigs? cores from live stems?).*

We will utilize the term "twig" henceforth.

*Lines 156-157 require more details because I do not know what they mean or how they apply to the Bayesian approach used. Figure 3 axes labels are unclear and/or inaccurate. Figure 7 has a polynomial fit but what order is that polynomial and why is it justified? What are the error bars in Figure 7B?  Figure 2 has no x-axis labels on some panels.*

Adding these details (such as the approach to the Bayesian mixing model) can easily be done in a revision.

*It is dubious for such problems to occur despite this paper having so many (native English speaking) authors approve its submission.*

We are not sure what is considered "dubious" here.  If the reviewer is alleging that the co-authors did not review the paper, then we would strongly contest the allegation.

*3) There are citation problems. Many of the citation choices are odd; for example, often papers are cited for ideas they mention as opposed what they actually demonstrate. E.g., Brooks et al., 2015 is overused and is used in odd places. Another odd one was Graup et al., 2022 (page 4 line 87), cited (I think) because Graup et al make a related assumptions; but, they do not show evidence to justify those assumptions. There are many of these throughout. Another common occurrence was confusing use of past citations when referring to the present study: e.g. "is a critical absence in terms of our capacity to close the transpiration budget (Cooper et al., 2020)" (213, page 8). How can a paper from 4 years ago be cited regarding this study's inability to close the balance? It needs to be clear why it is being cited. A similar issue arises again sentences later: "we estimate that aggregate hillslope water use was more similar between the conifers and aspens than implied from Figure 2 (Pataki et al., 2000). I do not know what the authors mean by this. There are many of these atypical citation usages in the discussion. The figures need to be more precisely cited in the text too; often figures would be referenced but it was unclear why.*

One way we could clarify the issues with citations is to be more explicit within the text as to why we chose certain citations.  For example, the reviewer brings up the Cooper et al., (2020) study.  We utilized this reference because it highlighted the importance of early spring water use by conifers in a similar montane system.  Because we did not have early spring sap velocity data for our sites, this was a useful paper to make inferences about the potential importance of water use prior to our annual installations of sensors. That said, the criticisms raised here are also rather odd.  For example, the reviewer penalizes us for using "past citations".  All citations are "past" – there are no future citations. We contest the idea that the citations usage here was truly "atypical".

*Despite the main conclusions relating to stand structure, very little literature on stand structure and its relationships with physiology, anatomy, and functional ecology is cited. The authors could greatly benefit from considering, for example, works by Hank Margolis and Jim Long on relationships between stand structure, sapwood area, and sapwood permeability that are problematically not considered in the interpretation of data shown in Figure 6.*

We will certainly do a better job of educating ourselves on missed citations such as those from Margolis and Long. Pointing out missed pieces of literature is a really critical part of the peer review process so we welcome this information.

*Statements such as "Much of the way we have come to understand the sensitivity of forests in the region to precipitation variability is through a handful of longer eddy covariance records (e.g. (Knowles et al., 2015)) and analysis of satellite greenness indices (e.g. (Trujillo et al., 2012))" are incorrect, as there have been decades of research on tree hydraulics using tree level or leaf level techniques to study drought responses in forests (also note that Trujillo et al did not focus on forests in this region!).*

We can rephrase this sentence and/or choose different citations. The point of this sentence was really about the primary methods we use to understand *temporal* changes in forest water fluxes in response to climate forcing, something which is difficult to infer from just hydraulic trait measurements.

*4) I am unclear how the authors justify the conclusions about interception and its relationship with stand structure.*

*It seems that inferences on interception are made from soil moisture data, and few details are provided on these soil moisture measurements. How many soil water measurements were used per plot? How do the soil moisture plots differ from the other study plots? How does density vary across the soil moisture plots? Is there enough spatial coverage by soil moisture sensors to rule out the possibility that the moisture differences are not random, or are not influenced by points where water drips from the canopy?*

One of the key changes we will make in the revision is more clearly articulating that interception is a *hypothesized* mechanism to help explain why forest stands may rely on water sources of different seasonal origins. We did not directly measure interception with precipitation gauges but from both the soil moisture data and the isotopic data, we have strong indirect evidence to support this idea. In the paper we present data from two soil moisture profiles taken from the instrumented hillslope. These two soil moisture profiles are in adjacent aspen and conifer stands. In addition, there is published data from a nearby hillslope in the same watershed (shown below from Carbone et al., (2022) but extended to the present) that show similar differences in the response of soil moisture to summer depending on the canopy. In addition, the isotopic data also show differences between species and between sites on the use of summer rain. Because root systems occupy a wider area than then a single soil moisture profile, this data is more realistic than representation of surface heterogeneity. In the revision, we can add the additional soil moisture data and be more explicit about potential heterogeneities with soil moisture.

[Figure]

*Figure 1: Difference in soil moisture (VWC) between adjacent aspen and conifer stands. The yellow streaks near surface capture moistening by summer rain under aspens.*

*More importantly, the choice of data presentation is highly concerning. Data were measured at 5 15 and 50 cm depths, and the values in Figure 5 show interpolations from that 5cm-50cm range. Given that the axis bounds are 5 (the upper y-axis limit) to 50 (the lower), essentially all of the range shown is influenced by the values at the 15-cm depth.*

We find ourselves a little confused by the reviewers concern here. Data was interpolated between the three soil moisture sensors (5, 15 and 50 cm). The bounds of the interpolations were set by the bounds of the measurements. In other words, we did not extrapolate beyond the depth we had measurements from. So, each of the three soil moisture probes contributed equally to the interpolation. For example, the soil moisture between 5 and 15 cm at each time step was estimated using a linear interpolation between these points. We could avoid the interpolation altogether and just show the data from the three depths if that would assuage this concern.

*It appears that the pattern being interpreted as moisture differences are mostly only a product of measurements at that one depth (15 cm). However, the y-axis seemingly uses a strange non-linear scale to emphasize the region of interpolated data most influenced by that 15 cm depth (15 +/- 10 makes up ~75% of the height of the figure!).*

We did use a log scale which was intended to mirror the quasi-logarithmic distribution of the sensors. The data show that under aspen stands, soil is drier at depth (50 cm) and moister near the surface (5 and 15 cm). Because 15 cm is the middle sensor, it does occupy more space in the figure but this depth is also specifically important because it is the depth where fine roots are likely to be densest.

*So, the blue patches in summer (Figure 5 c), critical to core arguments of this paper, may only be a product of measurements at 1 depth (15 cm), and effect is visually stretched to skew what readers might perceive. The axis scale and the interpolation approach together create a misleading picture.*

In the revision, we can use a monotonic color bar for panels b and c. However, the use of the brown-blue scale in Panel D emphasizes where soil moisture anomalies are positive vs.

negative.  This is a very typical approach when visualizing anomalies where the 0 point reflects a change in color.

*In conclusion: I think the findings could be presented in an entirely rewritten manuscript that might be acceptable for publication. There are interesting data and findings shown but their presentation would likely have to be as an entirely new manuscript because of the major issues cited above.*

It is always disappointing when a manuscript that is the product of extensive work by a large body of researchers resonates so poorly with a reviewer.  Many of the issues raised by the reviewer seem to stem from choices in language, presentation and a perceived sense of over-interpretation of results. We believe that with careful consideration in the writing – particularly in our scope of inference - and following the suggestions raised by Reviewer #2, we can address these issues while keeping the core results and analyses in the manuscript  intact. We feel the topics addressed here are important to the ecohydrological community, that the data and analyses are novel, and that the work will stimulate new studies. The dataset, while limited in many respects, is also extensive in terms that it covers multiple years of sap velocity over a high-elevation hillslope and includes 100's of isotopic measurements.

---

## Author Response (AR1)

**Response to Reviewer #1**

Thank you to Reviewer #1 for the constructive feedback on the manuscript. Detailed responses are listed below.

*Here are some possible overlooked aspects; the paper might benefit from addressing the long-term climate change implications on these dynamics in more detail, such as the potential for increased frequency of droughts or low snowpack years and its impact on forest ecosystems. Also, some crucial figures in this version seem to be a bit too small, making it difficult to read without zooming a lot.*

We have added some additional text in Lines 510-523 that discuss some possible projections arising from the work. We have adjusted the figures to deal with the sizing issues in the original submission.

*Comments by line.*

*Line 35, incomplete citation. (and more citations also inclmplete on the ms.)*

We have fixed the Strange et al. reference and checked for other errors.

*The hypothesis in line 90 is not fully clear to me. Does it mean that the nature of water availability will determine the stand sensitivity to changes in annual precipitation distribution? The second hypothesis made me think about the fact that canopy structure is a much more complex and dynamic outcome that depends on site conditions and not just water availability. Is having access to snowmelt indicative of deeper and better soil conditions?*

We have rewritten the end of the Introduction and removed the problematic hypotheses statements.

*Line 130 related to the deuterium correction might be a good idea to compare notes with https://hess.copernicus.org/articles/26/5835/2022/ that seem to relate this methodological "offset" with the volume of the water samples exposed to the CVD.*

We will have included discussion on the Diao et al., (2022) paper and discussed the volume of water we extracted from our samples.

*In methods, we have the seasonal origin index, which has not been introduced or has been introduced well enough.*

We removed all discussion and figures associated with the seasonal origin index (SOI). We really did not use or need this index in the discussion and it added confusion and unnecessary bulk to the paper.

*Figure 2: I recommend using one more color on the gradient of A, b, and C so the upper sap velocities >40 are easily distinguished between trees. Also, I would add the months to the figures. Not everyone is familiar with relating DOY and the months…*

We have adjusted the color bar to include multiple colors – it looks much better, thank you! We have also marked the month boundaries.

*Figure 4 could be larger for easy reading, and the dots of stem water might be better to plot with smaller dots and maybe somewhat transparent. (Just a suggestion). Also, what is the difference between SOI and the relative or reliance snowmelt use? Maybe it would be nice to make this clearer, as I am getting confused with the rest of the figures*

We have adjusted the dots on the figure to make them transparent and made the figure and text larger. As noted, we removed the SOI analysis, so the isotopic analysis is just based on the mixing model now.

*Line 250: I am not sure if I am following why there is reliance on seasonal rain if the snowmelt represents 80% of the annual.*

We have added text on Lines 312-317 to clarify this point.

*Line 365, maybe there is a bit of an offset between this and the message from the introduction. Maybe it would be good for the readers to be able to connect more clearly with this reminder on the discussion, and the canopy structure is supposed to be a key element of the manuscript, which was a bit in the shadows, in my opinion, throughout the manuscript also edit the citations years.*

In the Introduction on Lines 77-110, we have some extensive discussion on previous work regarding the influence of canopy structure on water use. These paragraphs provide a set up for the discussion and introduce some key papers on the topic (that we missed in the initial submission).

*Line 385 might be something to consider: reference the result sections to some figures or particular sections where this is supported instead of sending the readers to the supplementary info. It seems like this should be an important part of the manuscript.*

We have now brought figure on snowpack into the main body of the manuscript as Figure 8.

*On the supplemental material,*

*I would recommend adding a list of the figures in the first page.*

We have added a Table of Contents for the Supplement.

*FigS2 seems to be cut on the text above the figure also could be nice to make reference to the literature and the methods on how this was addressed.*

Figure S2 has been corrected to deal with the strange cropping.

*FigS3 here is something related to the SOI and relative contribution that was creating some confusion explained above, where it was not clear to me why the SOI is used in the main ms.*

As noted, we simply removed discussion of SOI to simplify the text.

*FigS6. I'm not sure if the y-axis labels are correct in both figures.*

The units have been adjusted to mm h-1, reported as "transpiration" and also changed so Panel A is mm h-1 and Panel B is difference in mm h-1 between years (i.e., presented as a "transpiration anomaly").

**Response to Reviewer #2**
We would like to thank Reviewer #2 for the detailed response and careful consideration of this manuscript. While we obviously see more merit to this work than the reviewer did, we acknowledge the important concerns they raised regarding the general inferences on ecohydrological processes we tried to draw from a relatively small observational dataset. We have made a substantial effort to reframe the paper around more modest inferences and also bring in broader set of papers on the effects of thinning that provide reference point for this work.

*1) The overall narrative and conclusions go beyond what should be inferred from the data. While the authors admit that the investigation was not established to pursue the narrative as it is presented, the truer statement is that the investigation falls short of what is needed to support the arguments regarding the role of canopy structure (which is reinforced by the scarcity of statistical tests).*

356-369 We appreciate the critical feedback raised by the reviewer, and we argue that some simple statistical analysis can support the idea that canopy structure is an important determinant of how trees respond to changing snowpack on this hillslope. Two of the stands we instrumented had similarly dense canopies, while the three other stands had similarly open stands. There was a total of 26 trees instrumented across these stands that had continuous sapflow measurements during 2019, 2021 and 2022.  Six of those trees were within the two dense stands, and *all* of those trees showed an increase in transpiration during the large snowpack year (2019).  There were 4 additional trees distributed across the three open stands that also showed a positive response.  The probability that all 6 trees in the two dense stands would show the same response just through random sorting is 0.002 based on a Monte Carlo simulation (we will include a similar analysis in the revision).  While this result does not prove that canopy structure *caused* the particular transpiration response to snowpack, it shows the response to changing snowpack was not random across the stands.  Secondly, the trees that responded positively to higher snowpack included all three species, including functionally distinct aspen, showing that the response is not specific to the traits or water-use strategy of one species. This analysis can be fleshed out in a revision but illustrates a probabilistic argument to support the finding that canopy structure was an important determinant of how trees responded to changing snowpack.

*Far more study sites with varying ecosystem structure would be needed to support the argument that topography shapes density, or that density relates to sensitivity, let alone to assert "the progression towards a low snowpack future will manifest at the sub-hillslope scale in dense stands". The story-telling narrative style involves of web of speculations, making it challenging to connect evidence with statements made throughout. A reasonable level of evidence is crucially unmet in many areas.*

As noted above, we acknowledge that landscape-scale projections from this limited dataset need to be treated with more caution.  We've added language such as Lines 432-436 that explicitly acknowledge the challenges posed by our network design and how this limits scaling. However, we'd also point out that Reviewer #1 came to an opposite conclusion i.e. they requested additional information about how these results could inform our understanding of how

montane forests will respond to climate change.  So, while we recognize the need to be cautious in not over extrapolating from the data, we also have posed some questions about (e.g. Lines 520-524) about how these results could guide future research on these questions.

*One key result is presented in line 281-283, "we see that the response of trees to the large spring snowpack of 2019 can best be predicted based on whether a given tree was located in one of the dense stands (i.e. where sapwood to ground area was ~38 cm2 m2) or open stands (i.e. sapwood to ground area ~12 m2 m2) (Figure 6)"; let's ignore the typo (it's not "m2 m2" but "cm^2 m^-2" ). Figure 6b shows 5 data points, and there is no aspen stand with low density and there is no mixed stand with high density, and thus there is no variation in density for a given composition of stand. This alone invalidates the primary argument of the paper (especially because we can see that species does matter in Figure 6a, Figure 4, and Figure 2). This critically small sample size does not allow for assessing stand effects.*

As discussed above, Lines 356-369 outline a simple statistical model to illustrate that the trees which responded positively vs. negatively to changes in snowpack were not randomly distributed across sites but clustered in a way that suggests canopy density might be a driver in the sensitivity to changing spring snowpack and (2) we see aspen, fir, and spruce trees responding differently whether they are in open vs. closed stands.  Without a full factorial design (i.e. dense aspen, open aspen, dense conifer, open conifer etc…) we cannot confidently conclude that the common response between the dense conifer and dense aspen were the result of the same mechanisms.  However, we have utilized the results from other thinning experiments to pose educated hypotheses such as the role of interception.

*2) There are major communications issues throughout, including use of imprecise language, omitted details, and insufficient copy editing.*

*Throughout, there is much suggestive and imprecise language, making it challenging to interpret and certainly not reproducible. Let us start with a few examples of uninterpretable statements:*

*-"we consolidate the numerous processes acting across sites by collapsing the measurement clusters…" (page 4 line 84)*

*-"it is also common to see areas where winter precipitation and groundwater may be of equal or greater importance to the species' water demands illustrating an overriding impact of hillslope context relative to species-specific traits" (line 64-65)*

*-"The sensitivity of annual tree activity to snow inputs did not clearly map onto species or elevational position on the hillslope." (line 280)*

*I do not know what these statements mean.*

We appreciate the reviewer pointing out examples of confusing language.  All these examples have been removed and/or rewritten.  In fact, we have added clarity to the text throughout the manuscript including use of more precise language.

*The imprecise language is even evident in the (arguably) most important sentences of the paper, with these two hypotheses: "Within this context, we first test two fundamental hypotheses: (1) the seasonal origin of water used by trees influences a stand's sensitivity to changes in precipitation seasonality and (2) that canopy structure is a reflection of the source of water accessible to the trees in that stand." I cannot understand how hypotheses of "influences" and "is a reflection" could be tested, establishing murkiness early on.*

As noted in response to Reviewer 1 above, we have removed these Hypotheses and rewritten the concluding paragraph of the Introduction (Lines 112-122).

*It is not just a writing-style issue because there are also major omissions and indicators of insufficient proofreading. Grammar problems occur throughout (e.g., a lack of appropriate punctuation).*

We carefully edited the revisions and hopefully these grammar problems have been removed. Without specific instances identified, it is difficult to be sure we have responded to all the concerns.

*Methodological details are too vague (although the Methods section is otherwise nicely structured). The sap flow study design is insufficiently communicated as it is unclear which trees and which trees in which sites were actually instrumented (Figure 1 only partially helps because there are places where the yellow circle could be over a deciduous or needleleaf tree, and there are issues such as only two yellow circles being visible in a site that supposedly had 4 to 6 trees instrumented).*

We appreciate the positive feedback on the Methods section structure. For the sap flow approach we have added more specifics regarding the sap velocity approach including the equations – see Lines 135-156. We apologies for the confusion the reviewer notes with Figure 1. We instrumented *2 trees per species per site* so for sites that were just aspen this meant just two trees were instrumented.  For sites that had all 3 species, this meant 6 trees. This is now explained on Lines 129-131

*It repeatedly says that 'live stems' were sampled for isotope data, which seems unfathomable for mature trees (were they live twigs? cores from live stems?).*

We will utilize the term "twig" henceforth and note approximate diameter of the twigs that were sampled (Line 175).

*Lines 156-157 require more details because I do not know what they mean or how they apply to the Bayesian approach used. Figure 3 axes labels are unclear and/or inaccurate. Figure 7 has a polynomial fit but what order is that polynomial and why is it justified? What are the error bars in Figure 7B?  Figure 2 has no x-axis labels on some panels.*

We have added more details to regarding the Bayesian mixing model on Lines 219-234.

*3) There are citation problems. Many of the citation choices are odd; for example, often papers are cited for ideas they mention as opposed what they actually demonstrate. E.g., Brooks et al., 2015 is overused and is used in odd places. Another odd one was Graup et al., 2022 (page 4*

*line 87), cited (I think) because Graup et al make a related assumptions; but, they do not show evidence to justify those assumptions. There are many of these throughout. Another common occurrence was confusing use of past citations when referring to the present study: e.g. "is a critical absence in terms of our capacity to close the transpiration budget (Cooper et al., 2020)" (213, page 8). How can a paper from 4 years ago be cited regarding this study's inability to close the balance? It needs to be clear why it is being cited. A similar issue arises again sentences later: "we estimate that aggregate hillslope water use was more similar between the conifers and aspens than implied from Figure 2 (Pataki et al., 2000). I do not know what the authors mean by this. There are many of these atypical citation usages in the discussion. The figures need to be more precisely cited in the text too; often figures would be referenced but it was unclear why.*

We have made a number of changes to make the use of particular references more transparent. For example, the reviewer noted confusion with respect to the Cooper et al reference. We now write: *"Previous sap velocity work from a subalpine system in the Sierra Nevada of California also suggest that transpiration is active by April and can reach nearly peak values by mid to late May (Cooper et al., 2020)."*

*Despite the main conclusions relating to stand structure, very little literature on stand structure and its relationships with physiology, anatomy, and functional ecology is cited. The authors could greatly benefit from considering, for example, works by Hank Margolis and Jim Long on relationships between stand structure, sapwood area, and sapwood permeability that are problematically not considered in the interpretation of data shown in Figure 6.*

We have expanded the references and discussion around stand structure with a number of key papers including: O'Donnell et al., 2021, Belmonte et al., 2022, Bréda et al. (1995) Kerhoulas et al. (2013), Fernandes et al. (2016) and Sohn et al. (2014). These paper all provide critical perspective on the way that stand structure influences water use and seasonality of water sources.

*Statements such as "Much of the way we have come to understand the sensitivity of forests in the region to precipitation variability is through a handful of longer eddy covariance records (e.g. (Knowles et al., 2015)) and analysis of satellite greenness indices (e.g. (Trujillo et al., 2012))" are incorrect, as there have been decades of research on tree hydraulics using tree level or leaf level techniques to study drought responses in forests (also note that Trujillo et al did not focus on forests in this region!).*

We have removed this sentence.

*4) I am unclear how the authors justify the conclusions about interception and its relationship with stand structure.*

*It seems that inferences on interception are made from soil moisture data, and few details are provided on these soil moisture measurements. How many soil water measurements were used per plot? How do the soil moisture plots differ from the other study plots? How does density vary across the soil moisture plots? Is there enough spatial coverage by soil moisture sensors to rule out the possibility that the moisture differences are not random, or are not influenced by points where water drips from the canopy?*

We have now shifted the discussion such that interception is proposed as one *hypothesized* mechanism to help explain why forest stands may rely on water sources of different seasonal origins.  We did not directly measure interception with precipitation gauges but from both the soil moisture data and the isotopic data, we have strong indirect evidence to support this idea.  In the paper we present data from two soil moisture profiles taken from the instrumented hillslope.  These two soil moisture profiles are in adjacent aspen and conifer stands (Figure 5).  In addition, there is published data from a nearby hillslope in the same watershed shown in Figure S6 that illustrates a similar dynamic.  In addition, the isotopic data also show differences between species and between sites on the use of summer rain.  Because root systems occupy a wider area than then a single soil moisture profile, this data is more realistic than representation of surface heterogeneity.

*More importantly, the choice of data presentation is highly concerning. Data were measured at 5 15 and 50 cm depths, and the values in Figure 5 show interpolations from that 5cm-50cm range. Given that the axis bounds are 5 (the upper y-axis limit) to 50 (the lower), essentially all of the range shown is influenced by the values at the 15-cm depth.*

We find ourselves a little confused by the reviewers concern here.  Data was interpolated between the three soil moisture sensors (5, 15 and 50 cm) in Figure 5 and Figure S6.  The bounds of the interpolations were set by the bounds of the measurements. In other words, we did not extrapolate beyond the depth we had measurements from. So, each of the three soil moisture probes contributed equally to the interpolation. For example, the soil moisture between 5 and 15 cm at each time step was estimated using a linear interpolation between these points.

*It appears that the pattern being interpreted as moisture differences are mostly only a product of measurements at that one depth (15 cm). However, the y-axis seemingly uses a strange non-linear scale to emphasize the region of interpolated data most influenced by that 15 cm depth (15 +/- 10 makes up ~75% of the height of the figure!).*

We did use a log scale in Figure 5 which was intended to mirror that the sensors are more focused near the surface than at depth.  The data show that under aspen stands, soil is drier at depth (50 cm) and moister near the surface (5 and 15 cm).  Because 15 cm is the middle sensor, it does occupy more space in the figure but this depth is also specifically important because it is the depth where fine roots are likely to be densest.

*So, the blue patches in summer (Figure 5 c), critical to core arguments of this paper, may only be a product of measurements at 1 depth (15 cm), and effect is visually stretched to skew what readers might perceive. The axis scale and the interpolation approach together create a misleading picture.*

The use of the brown-blue scale in Panel D of Figure 5 emphasizes where soil moisture anomalies are positive vs. negative.  This is a very typical approach when visualizing anomalies where the 0 point reflects a change in color.

*In conclusion: I think the findings could be presented in an entirely rewritten manuscript that might be acceptable for publication. There are interesting data and findings shown but their presentation would likely have to be as an entirely new manuscript because of the major issues cited above.*

Thank you again to Reviewer 2 for the careful consideration and concerns raised here.  In reading this review, we appreciate the ways the initial presentation may have been overly confident and, in places, confusing.  The revision intended to strike a more modest tone while still emphasizing the way conclusions from this study could inform future studies. We have also sought out simpler language choices and drew information from a wider literature. The data presented here is important despite limitations from the experimental design and we hope the reviewer is able to appreciate that from the revised text.

---

## Author Response (AR2)

Dear Editor,

On behalf on myself and co-authors, I'd like to thank you and the reviewers for assistance with the manuscript. We have made changes to the manuscript to address the small issues raised by Reviewers 1 and 2. Responses to the specific comments are addressed below.

Best regards,
Max Berkelhammer

**Reviewer 1:**

Check-In Figure 4, the color of the twig water needs to be more evident, and it gets confused with the color of the snowmelt with low values.

We have added a subpanel to this figure that separates the precipitation from the xylem water so both can be easily seen without blocking each other.

Check-I would suggest editing the result section to explain the isotopic data in section 3.2 (the first paragraph, in particular, is hard to follow), where the water partitioning from snow, precipitation stream flow, and transpiration are quickly explained.

We have added details to the first paragraph of Section 3.2 to increase explanation of the results.

**Reviewer 2:**

Based on the comprehensive revisions and the authors' responsiveness to feedback, I recommend that the manuscript be accepted for publication after minor revisions. The suggested improvements will enhance the manuscript's clarity and impact, ensuring it effectively communicates its significant contributions to the field. I appreciate the authors' diligent efforts in revising the manuscript. The study offers valuable insights into how canopy structure influences ecohydrological processes, particularly in the context of snowmelt utilization and climate variability.

We appreciate the positive feedback on the work.

   i.)    the influence of legacy-effects: trees do not always react immediately and stress or recovery-effects might show up delayed und decoupled from year-to-year variations;

We have added discussion in Lines 519-523 to bring up questions on the role that legacy may be playing in the response to variations in seasonal precipitation inputs.

   ii.)    ii.) the role of differing rooting patterns between thinned and dense stands: while contrasting literature is cited in the introduction, this highly interesting aspect is not touched later. In dense stands, deeper rooting patterns might develop due to

competition, and thinned stands might develop shallower rooting systems (see Schenk 2022 the shallowest possible rooting system). The explained differences in interception might also affect root distribution and, hence, the dependence on precipitation/deeper water. If the soil water isotope data would be higher resolved (and not only from the first 10 cm), the mixing model could've used in a much more effective way, in my opinion (i.e., by estimating water uptake depths of the trees).

We have added reference to Schenk 2008 and more extensive discussion on root water uptake profiles throughout the manuscript.

iii.)    iii.) the influence of snowmelt running off from the slope and being more available downslope (i.e., snowmelt could be more important for trees located downstream).

In lines 70-75 and elsewhere we discussed the Martin et al., 2018 paper that explicitly addresses the importance of downstream snowmelt flow.

- l. 14, l.47/48: Interception or rooting depth? (see main comments)

We added text to discuss the role of both above and belowground processes.

- l. 97-99: the statement contradicts what is said in l. 85-87, which is imo a good starting point for the ms - but later it is not referred to this interesting aspect

We used some text to try and reconcile the apparent contradictions.

chdck - l.119: "difference in water resources" - imprecise, please be specific what is meant here

We removed the reference to water resources because it was a little confusing and unnecessary.

- l. 119/120: total transpiration demand of dense stands will be higher than for thinned stands - could this alone explain the results obtained?

Here and a few places we noted the higher water demands of these dense stands as part of the reason they are more able to rely on big snowmelt years..

check - l.192: "small samples" - imprecise, "small amounts of sample material"?

We specify this value to be 0.6 ml.